# 🦉OWL: Optimized Workforce Learning
## *General Multi-Agent Assistance for Real-World Task Automation*

**Mengkang Hu**[1,4*], **Yuhang Zhou**[1*], **Wendong Fan**[2], **Yuzhou Nie**[3], **Bowei Xia**[1],
**Tao Sun**[2], **Ziyu Ye**[5], **Zhaoxuan Jin**[6,2], **Yingru Li**[7], **Qiguang Chen**[1], **Zeyu Zhang**[8,2],
**Yifeng Wang**[4], **Qianshuo Ye**[9,2], **Bernard Ghanem**[10], **Ping Luo**[1†], **Guohao Li**[2†]

[1] The University of Hong Kong    [2] Eigent.AI    [3] UCSB    [4] CAMEL-AI
[5] The University of Chicago    [6] Northwestern University    [7] CUHK
[8] ANU    [9] University of Cambridge    [10] KAUST

## Abstract

Large Language Model (LLM)-based multi-agent systems show promise for automating real-world tasks but struggle to transfer across domains due to their domain-specific nature. Current approaches face two critical shortcomings: they require complete architectural redesign and full retraining of all components when applied to new domains. We introduce WORKFORCE, a hierarchical multi-agent framework that decouples strategic planning from specialized execution through a modular architecture comprising: *(i)* a *domain-agnostic* **Planner** for task decomposition, *(ii)* a **Coordinator** for subtask management, and *(iii)* specialized **Workers** with *domain-specific* tool-calling capabilities. This decoupling enables cross-domain transferability during both inference and training phases: During inference, WORKFORCE seamlessly adapts to new domains by adding or modifying worker agents; For training, we introduce OPTIMIZED WORKFORCE LEARNING (OWL), which improves generalization across domains by optimizing a domain-agnostic planner with reinforcement learning from real-world feedback. To validate our approach, we evaluate WORKFORCE on the GAIA benchmark, covering various realistic, multi-domain agentic tasks. Experimental results demonstrate WORKFORCE achieves open-source state-of-the-art performance (**69.70%**), outperforming commercial systems like OpenAI's Deep Research by **2.34%**. More notably, our OWL-trained 32B model achieves **52.73%** accuracy (**+16.37%**) and demonstrates performance comparable to GPT-4o on challenging tasks. To summarize, by enabling scalable generalization and modular domain transfer, our work establishes a foundation for the next generation of general-purpose AI assistants. [1]

## 1 Introduction

Large Language Models (LLMs) have undergone a period of rapid advancement, evolving from simple text predictors into powerful autonomous agents capable of planning, tool use, and multi-step reasoning [OpenAI, 2024, Anthropic, 2025]. Recently, multi-agent systems (MAS) have emerged as a promising approach for complex real-world tasks, demonstrating that dividing tasks among specialized agents can enhance performance [Li et al., 2023, Qian et al., 2023, Wang et al., 2024].

Although current MAS have yielded impressive results, their designs are typically domain-specific, severely restricting **cross-domain transferability**. This shortcoming appears in two forms: *(i)*

---

39th Conference on Neural Information Processing Systems (NeurIPS 2025).

[1]Our code is available at https://github.com/camel-ai/owl.

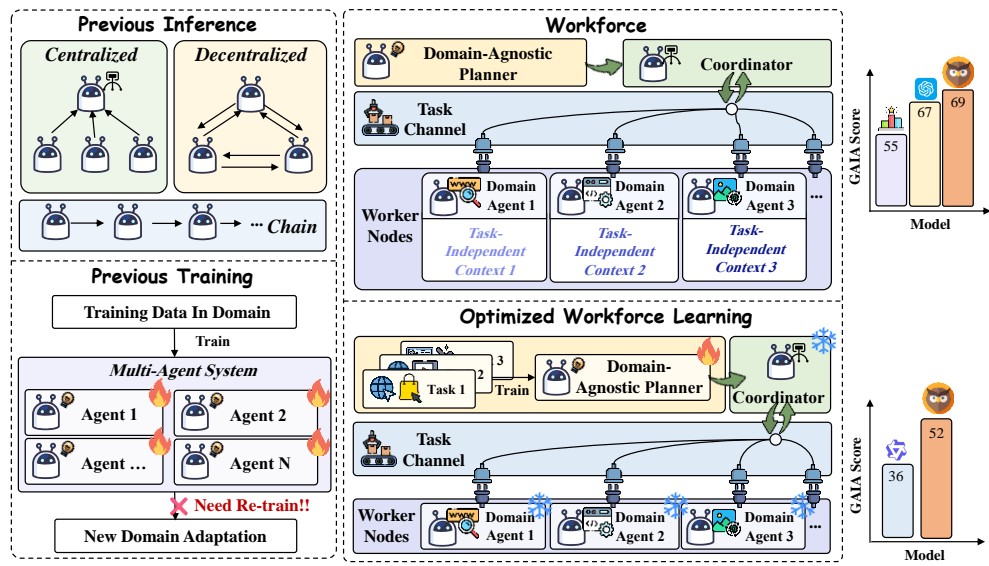

Figure 1: Overview of WORKFORCE and OPTIMIZED WORKFORCE LEARNING.

First, on the **inference** side, deploying a system in a new domain often entails a full redesign; for instance, MetaGPT [Hong et al., 2023] depends on Standard Operating Procedures tailored to software engineering, hindering its extension to other fields. *(ii)* Second, on the **training** side, existing works often optimize every agent. MALT [Motwani et al., 2024], for example, follows a fixed generator-verifier-refiner pipeline, necessitating separate training for every component. Consequently, migrating such systems demands retraining the entire agent ensemble, sharply reducing flexibility. These drawbacks underscore the need for a generalized, modular multi-agent architecture that can be rapidly adapted to diverse domains with minimal retraining and redesign.

Building upon these observations, firstly, we introduce WORKFORCE, a hierarchical **multi-agent inference** framework that decouples strategic planning from domain-specific execution. As is shown in Figure 1, This modular design comprises three core components: *(i)* Domain-agnostic **Planner**: Generates abstract task decompositions based on high-level goals. *(ii)* **Coordinator**: Assigns subtasks to appropriate workers. *(ii)* Domain-specific **Worker Nodes**: A set of specialized agents that perform tool calls to accomplish each subtasks. The decoupling of these components enables plug-and-play extensibility, allowing WORKFORCE to be seamlessly adapted to new domains by simply replacing or adding worker nodes. Furthermore, this modular architecture facilitates OPTIMIZED WORKFORCE LEARNING (OWL), a novel **multi-agent training** paradigm. OWL focuses on enhancing MAS's cross-domain transferability by training a generalizable domain-agnostic planner. Specifically, we employ a two-stage training strategy: Supervised Fine-Tuning (SFT) for planner initialization, followed by Reinforcement Learning [Rafailov et al., 2023] to further enhance the model's generalization capabilities.

We evaluate our approach on the GAIA benchmark, a rigorous suite for generalist AI assistants that spans diverse domains and demands multimodal reasoning, code execution, and live web search [Mialon et al., 2023]. WORKFORCE attains an accuracy of **69.70%**, surpassing strong commercial proprietary baselines like OpenAI's Deep Research [OpenAI, 2024] (55.15%). To demonstrate the effectiveness of OWL, we further post-trained a strategic planner initialized by Qwen2.5-32B-Instruct [Yang et al., 2024] on a custom-curated training dataset without using any GAIA data. After training, the model reaches 52.73% score (**+16.37%**), outperforming models like GPT-4o-mini [OpenAI, 2024] (47.27%) and Qwen2.5-72B-Instruct (49.09%). These results confirm that our modular training strategy generalizes across domains while requiring minimal retraining.

**Our key contributions are fourfold:**

**1. A New Flexible and Modular Multi-Agent Architecture.** We propose WORKFORCE that is modular and scalable in both inference and training, enhancing cross-domain transferability.

**2. State-of-the-Art Performance.** Our system achieves open-source state-of-the-art performance on the GAIA benchmark, even surpassing proprietary systems like OpenAI's Deep Research.

**3. Efficient and Effective Training Paradigm.** OWL significantly enhances model capabilities with minimal overhead, enabling Qwen2.5-32B-Instruct to achieve a **16.37%** performance gain and reach comparable performance to GPT-4o on challenging tasks.

**4. Fully Open-Source.** We release all code, models, and data to support open research.

## 2 Preliminary

**Large Language Model (LLM)-based Agents** are autonomous systems that perceive, reason, and act in various environments [Xie et al., 2023]. These agents operate in a perception-reasoning-action loop, where they observe their environment, process information through the language model, determine appropriate actions, and execute them to achieve goals.

**Multi-Agent Systems** extend this paradigm by enabling multiple LLM-based agents to collaborate on complex tasks. Frameworks such as CAMEL [Li et al., 2023] and MetaGPT [Hong et al., 2023] have demonstrated that collaborative approaches can outperform single-agent systems on tasks requiring diverse expertise. However, existing multi-agent frameworks are typically constrained by domain-specific designs that limit their broader applicability. This paper aims to develop scalable and generalizable multi-agent frameworks WORKFORCE through strategic planning and task execution decoupling, enabling efficient coordination across diverse domains. (See complete formalization in Appendix C and more related works in Section 6.)

**Generalist AI Assistant** was first introduced by GAIA [Mialon et al., 2023]. These systems are designed to handle a diverse range of complicated tasks across multiple domains and modalities. As the first question answering benchmark evaluating generalist AI assistants, GAIA proposes to enable LLM-based agents to gather information in real-world contexts, testing fundamental capabilities including multi-modal understanding, web browsing, reasoning, and complex problem-solving. Recently, numerous companies have released generalist AI assistant products (e.g., OpenAI's Deep Research [OpenAI, 2024]). While open-source frameworks have made significant progress (e.g., Huggingface's Open Deep Research [Roucher et al., 2025a]), they still lag behind commercial solutions. In this paper, we aim to close the gap between open-source and commercial proprietary agent frameworks. The proposed WORKFORCE surpasses OpenAI's Deep Research by 2.34%, while our training method OWL significantly improves Qwen2.5-32B-Instruct's performance by 16.37%.

## 3 Multi-Agent Inference: WORKFORCE

### 3.1 WORKFORCE

*Motivation.* Contemporary multi-agent systems are critically limited by domain specificity and architectural rigidity, requiring complete redesign and retraining for each new application domain. We introduce WORKFORCE, which addresses this fundamental challenge through modular architecture, particularly the separation of domain-agnostic planning from domain specific execution. More details about WORKFORCE can be found in Appendix D.

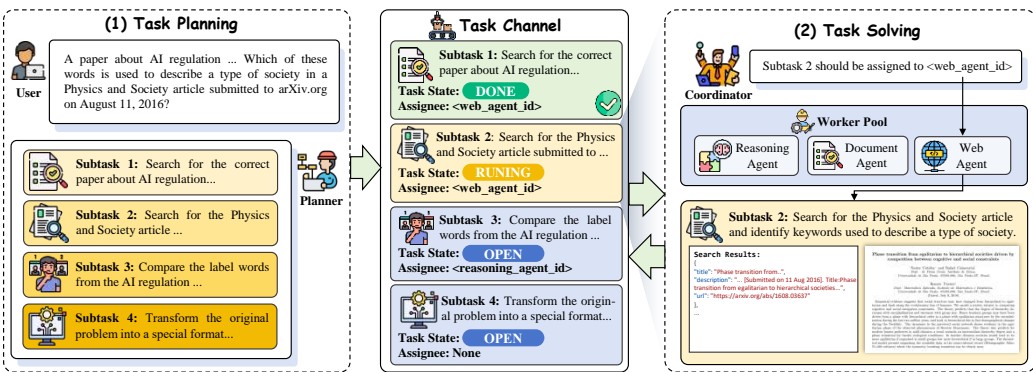

Figure 2: Overview of the WORKFORCE framework. The system consists of a Planner Agent for task decomposition, a Coordinator Agent for orchestrating subtasks, and multiple specialized Worker Nodes equipped with domain-specific toolkits to execute assigned tasks.

*Architecture.* As illustrated in Figure 2, WORKFORCE comprises three core components: *(i)* **Planner Agent** analyzes incoming tasks and decomposes them into subtasks based on worker capability registry; *(ii)* **Coordinator Agent** serves as the central orchestration mechanism, managing worker assignments and task dependencies while integrating intermediate results; *(iii)* **Worker Nodes** consist of one or more specialized agents equipped with specific capabilities and toolkits that execute assigned subtasks and post results. This modular architecture provides inherent flexibility, allowing the framework to be deployed across diverse applications by simply modifying the worker nodes while maintaining its core planning and coordinating mechanism.

*Communication Mechanism.* Communication in WORKFORCE operates through a shared *task channel* that serves as a central hub. The coordinator posts tasks and assignments to this channel. Upon completion, workers post only their final results back to the channel, while the detailed execution context of tool calls remains isolated within each subtask's scope. This maintains a clean context for each worker, who only has access to the current subtask details and concise previous subtask results. This centralized approach simplifies system management and enhances scalability by eliminating direct agent-to-agent messaging.

*Task Flow.* The task processing workflow within WORKFORCE follows a structured pipeline: *(i)* The Planner analyzes incoming overall tasks and then decomposes them into a set of subtasks according to the capabilities of available worker nodes and the overall task's complexity; *(ii)* The Coordinator assesses the capabilities of available worker nodes and dispatches subtasks accordingly; *(iii)* The Worker Nodes perform the assigned subtasks using their specialized tools; *(iv)* Results from worker nodes are posted to the shared *task channel*; *(v)* The Coordinator manages task dependencies and integrates results, ultimately forwarding them to the Planner; *(vi)* The Planner analyzes the results of each subtask and synthesizes the final output.

*Replanning Mechanism.* During task execution, workers self-assess whether the assigned subtask is failed. When a worker determines that a subtask has failed, it posts failure information to the task channel. The task channel then detects this failure, and the planner will be prompted to generate new subtasks based on the feedback information. This replanning mechanism enables test-time scaling by dynamically adjust its approach to increasingly complex tasks, which is validated in Section 5.

## 3.2 Generalist Multi-Agent Assistance

To construct a generalist multi-agent assistant capable of processing diverse real-world tasks, we instantiated WORKFORCE with three worker agents, each equipped with domain-specific toolkits: *(i)* A **Web Agent**, capable of performing web searches, extracting webpage content, and simulating browser actions; *(ii)* A **Document Processing Agent**, designed to process documents and multimodal data, including text, images, audio, video, spreadsheets, and various file formats; and *(iii)* A **Reasoning/Coding Agent**, which handles analytical reasoning and code execution tasks. The details of worker agents and corresponding toolkits could be found in Appendix D.3.

## 3.3 Experiments

*Baselines.* We selected a comprehensive set of baseline systems, categorized into four main groups: *(i)* **Proprietary frameworks** establish upper-bound commercial performance, including commercial agentic systems like OpenAI's Deep Research [OpenAI, 2024], h2oGPTe Agent [H2O.ai, 2024], etc. *(ii)* **Open-source frameworks** reveal the community progress, including strong baselines like HuggingFace's Open Deep Research [Roucher et al., 2025a], Trase Agent [Trase, 2024], etc. *(iii)* A **Single Agent** baseline utilizing multi-step tool calling. *(iv)* **Role Playing** Li et al. [2023] comprising two agents (user agent and assistant agent) that collaborate through structured dialogue to accomplish tasks. Note that to control experimental variables, both the Single Agent and Role Playing baselines utilize identical tool sets to WORKFORCE.

*Implementation Details.* For our implementation, we access all models via APIs, eliminating the need for GPUs. To ensure reproducibility, we configure API inference with greedy decoding. The default replanning threshold is set to 2. For evaluation methodology, we employ pass@3 sampling for WORKFORCE with GPT-4o [OpenAI, 2024] and pass@1 for WORKFORCE with Claude-3.7-sonnet [Anthropic, 2025]. Since some gold answers from GAIA had been leaked online, we blocked several websites to ensure fair comparison [OpenAI, 2024].

Table 1: Performance comparison of agent frameworks on GAIA [Mialon et al., 2023] validation set with accuracy score (%) as the evaluation metric. Scores of open-source and proprietary frameworks were obtained from the official leaderboard. The best-performing proprietary and open-source frameworks are highlighted in **bold**.

| Agent Name | Base Model | Level 1 | Level 2 | Level 3 | Average |
|---|---|---|---|---|---|
| *Proprietary Frameworks* | | | | | |
| DRP-val-v.1.0 | - | 56.60 | 48.84 | 15.38 | 46.06 |
| omne | O1-Preview[†] | 60.38 | 44.19 | 23.08 | 46.06 |
| Barcelona v0.1 | Claude-3.5-Sonnet[†] | 62.26 | 50.00 | 26.92 | 50.30 |
| Ormind v0.1 | Claude[†] | 69.81 | 54.65 | 26.92 | 55.15 |
| desearch | GPT-4o | 71.70 | 58.14 | 23.08 | 56.97 |
| Anges | Claude[†] | 66.04 | 65.12 | 30.77 | 60.00 |
| h2oGPTe Agent v1.6.8 [H2O.ai, 2024] | Claude-3.5-Sonnet | 67.92 | 67.44 | 42.31 | 63.64 |
| Deep Research [OpenAI, 2024] | O3 | 78.66 | **73.21** | **58.03** | 67.36 |
| Trase Agent v0.3 [Trase, 2024] | Claude[†] | **83.02** | 69.77 | 46.15 | 70.30 |
| Langfun Agent v2.1* [Google, 2023] | Claude-3.7-Sonnet[†] | **83.02** | 68.60 | 57.69 | **71.52** |
| *Open-Source Frameworks* | | | | | |
| FRIDAY [Wu et al., 2024] | GPT-4-Turbo | 45.28 | 34.88 | 11.54 | 34.55 |
| Multi-Agent Exp v0.1 [Microsoft, 2024] | GPT-4-Turbo | 54.72 | 38.37 | 11.54 | 39.39 |
| HuggingFace Agents [Roucher et al., 2025b] | GPT-4o | 58.49 | 43.02 | 19.23 | 44.24 |
| Magnetic-One [Fourney et al., 2024a] | O1[†] | 56.60 | 46.51 | 23.08 | 46.06 |
| AutoAgent [Tang et al., 2025] | Claude-3.5-Sonnet | 71.70 | 53.49 | 26.92 | 55.15 |
| Open Deep Research [Roucher et al., 2025a] | O1 | 67.92 | 53.49 | 34.62 | 55.15 |
| TapeAgents [Bahdanau et al., 2024] | Claude-3.7-Sonnet | 71.70 | 53.49 | 30.77 | 55.76 |
| Single Agent | GPT-4o | 52.83 | 34.88 | 15.38 | 37.58 |
| Role Playing [Li et al., 2023] | GPT-4o | 75.47 | 52.33 | 19.23 | 54.55 |
| Role Playing[‡] [Li et al., 2023] | GPT-4o[†] | 81.14 | 54.65 | 23.08 | 58.18 |
| *Ours* | | | | | |
| WORKFORCE | GPT-4o | 81.14 | 58.14 | 26.92 | 60.61 |
| WORKFORCE | Claude-3.7-Sonnet | **84.91** | **68.60** | **42.31** | **69.70** |

***Main Results.*** Several conclusions could be drawn from Table 1:

***(i)* Workforce achieves state-of-the-art performance among open-source frameworks.** Our WORKFORCE achieves **69.70%**, consistently outperforming previous open-source frameworks across all difficulty levels. Under strictly controlled settings with the same model and toolkits, our GPT-4o-based WORKFORCE achieves **60.61%** accuracy - **23.03%** higher than the Single Agent and **6.06%** higher than the multi-agent baseline Role Playing.

***(ii)* Workforce demonstrates comparable or even superior performance relative to commercial proprietary frameworks.** While previous open-source frameworks have exhibited a substantial performance gap compared to closed-source alternatives, WORKFORCE considerably narrows this divide. To our knowledge, WORKFORCE is the first open-source system to exceed OpenAI's Deep Research, achieving a **2.34%** improvement, and also establishes a new Level 1 best by outperforming Langfun Agent v2.1 by **1.89%**.

## 4 Multi-Agent Training: OPTIMIZED WORKFORCE LEARNING

### 4.1 Training Strategy

***Motivation.*** The WORKFORCE architecture separates domain-agnostic planning from domain-specific execution, allowing us to adapt to new domains by simply adding or replacing Worker Nodes while preserving the core planning mechanism. We introduce **OPTIMIZED WORKFORCE LEARNING (OWL)**, which focuses on enhancing a generalizable Planner Agent capable of handling diverse

---

[1][†] Indicates systems that use multiple models, and only the primary model is listed.
[*] Indicates frameworks that have released an early version but have not provided the latest version that can replicate these results.
[‡] Indicates an optimized version of role-playing where we introduced an LLM-based classifier that switches to O3-mini when tasks require more reasoning and coding, and uses GPT-4o otherwise.

Table 2: The statistics of the dataset. We use icons to represent different agent capabilities:🖥️Web Browsing; 🧠Reasoning; 📄Document Processing; 🖥️Coding; 🖼️Multimodal Handling. More details of training data can be found in Appendix D.4.

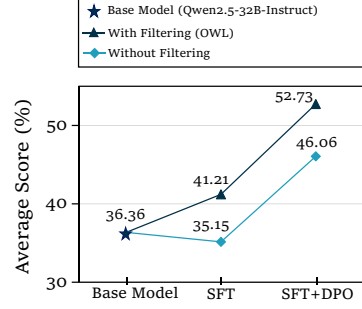

| Dataset | # Total | # SFT Filtering | # DPO Filtering | Agent Capabilities |
|---|---|---|---|---|
| HotpotQA | 998 | 495 | 354 | 🖥️ 🧠 🖼️ |
| WikiTableQuestions | 869 | 577 | 311 | 📄 🧠 |
| Math-related | 1100 | 487 | 297 | 🧠 🖥️ |
| Infinity-MM | 499 | 40 | 47 | 🧠 🖼️ |
| Total/Average | 3466 | 1599 | 1009 | |

(a) Ablation study on trajectory filtering

real-world scenarios. This design significantly reduces training overhead since only the Planner Agent requires intensive optimization, while Worker Nodes can leverage existing domain-specific tools with minimal adaptation. This "stable core, variable periphery" approach enables efficient knowledge transfer across domains and eliminates the need to retrain entire systems for new applications, substantially reducing computational costs while maintaining consistent performance.

***Implementation.*** More specifically, we adopt a two-phase training paradigm: *(i)* In the first phase, we employ supervised fine-tuning (SFT) to initialize the Planner Agent with fundamental task decomposition skills derived from expert demonstrations. *(ii)* Subsequently, we utilize reinforcement learning to further optimize the SFT-initialized Planner Agent. We select Direct Preference Optimization (DPO) [Rafailov et al., 2023] as our optimization algorithm of choice, as this phase enhances the quality of decomposition strategies beyond mere imitation of demonstrations, enabling the Planner to develop more sophisticated decision-making capabilities.

## 4.2 Task Curriculum

***Motivation.*** The core innovation of WORKFORCE is its architectural separation of domain-agnostic planning from domain-specific execution. For this design to be effective, the Planner Agent must possess robust generalization capabilities across diverse problem domains. This creates a fundamental tension: the Planner must simultaneously maintain a deep understanding across disparate domains while avoiding overfitting to specific task patterns or domains. To address this challenge, we developed a strategically balanced task curriculum that deliberately spans multiple capability dimensions required for general intelligence. Our curriculum design is guided by two key principles: *(i)* **Capability Coverage**: Exposing the Planner to diverse reasoning patterns and problem structures. *(ii)* **Transfer Learning**: Prioritizing tasks that develop complementary cognitive skills which can transfer across domains, rather than domain-specific knowledge.

***Implementation.*** More specifically, as is shown in Table 2, we carefully selected four datasets, each targeting distinct agent cognitive capabilities dimensions: *(i)* **HotpotQA** [Yang et al., 2018]: This dataset requires multi-hop reasoning based on online information, challenging the Planner to orchestrate complex information-seeking behaviors. *(ii)* **WikiTableQuestions** [Pasupat and Liang, 2015]: This dataset requires the Planner to formulate strategies for navigating, filtering, and operating over tabular information. *(iii)* **Math-related Problems**: This is a custom-curated collection of mathematical problems that require reasoning or coding to solve, covering various mathematical domains. They help the Planner develop skills in logical reasoning and computational problem-solving. *(iv)* **Infinity-MM** [Gu et al., 2024]: As a multimodal dataset, Infinity-MM challenges the Planner to orchestrate multimodal information processing., including vision, text, and structured data.

## 4.3 Trajectory Synthesis

***Supervised Fine-tuning.*** We employed our WORKFORCE approach (§3.1) with GPT-4o-mini to synthesize expert trajectories, which consist of planner-generated subtasks and worker-generated execution traces. To filter out low-quality data, we applied different evaluation metrics across datasets: For HotpotQA and WikiTableQuestions, we used accuracy metrics; For Infinity-MM, we used text cosine similarity with a 0.7 threshold between ground truth and generated answers; For math-related problems, we implemented LLM-as-a-judge to compare ground truth answers with workforce-generated solutions. Ultimately, as shown in Table 2 and 6, we obtained 1,599 filtered trajectories for supervised fine-tuning, with each trajectory containing an average of 3.41 subtasks.

Table 3: Performance comparison between different planners. Each result is based on the Workforce framework with `GPT-4o` as the worker foundation model.

| Planner | Level 1 | Level 2 | Level 3 | Average |
|---|---|---|---|---|
| GPT-4o-mini [OpenAI, 2024] | 64.15 | 45.34 | 19.23 | 47.27 |
| Qwen2.5-72B-Instruct [Yang et al., 2024] | 60.30 | 51.16 | 19.23 | 49.09 |
| Claude-3.7-Sonnet [Anthropic, 2025] | 81.13 | 53.49 | 34.61 | 59.39 |
| GPT-4o [OpenAI, 2024] | 81.14 | 58.14 | 26.92 | 60.61 |
| Qwen2.5-32B-Instruct [Yang et al., 2024] | 49.05 | 33.72 | 19.23 | 36.36 |
| *w.* OWL (SFT) | 56.60 +7.55 | 39.53 +5.81 | 15.38 -3.85 | 41.21 +4.85 |
| *w.* OWL (SFT + DPO) | 67.92 +18.87 | 51.16 +17.44 | 26.92 +7.69 | 52.73 +16.37 |

***Reinforcement Learning.*** We use the SFT-initialized model to generate pair-wise trajectories for DPO [Rafailov et al., 2023]. Specifically, for each question in our collected dataset, we roll out $n = 4$ distinct trajectories. The evaluation of these generated trajectories is the same as in the SFT stage. Then, we construct preference pairs from the $n$ trajectories generated for each question based on their evaluation outcomes. For Math, HotpotQA, and WikiTableQuestions tasks, trajectories that are correct are labeled as "chosen", while incorrect answers are labeled as "rejected". For Infinity-MM dataset, trajectories whose final text cosine similarity scores exceeded the same threshold as in the SFT stage (0.7) are labeled as "chosen", while those below the threshold are labeled as "rejected". As shown in Table 2, we collect 1009 filtered trajectories pairs.

## 4.4 Experiments

***Baselines.*** We compare our approach against multiple proprietary and open-source models as baselines, including the GPT-4o series [OpenAI, 2024], Claude-3.7-Sonnet [Anthropic, 2025], and the Qwen2.5 series [Yang et al., 2024]. These models represent the current state-of-the-art in language model capabilities across different scales and architectures.

***Implementation Details.*** Our model training is conducted on a computing cluster equipped with 8 NVIDIA H100 GPUs. We use the LlamaFactory [Zheng et al., 2024] framework for managing and executing our training procedures. Specifically, for all the models we trained, the input sequences are truncated to a maximum length of 32,768 tokens and the learning rate is set to $10^{-5}$. All models are trained for a total of two epochs. To optimize memory usage and training efficiency, we use bfloat16 mixed-precision training. The effective batch size is 12, achieved through 1 per-device batch size combined with 12 gradient accumulation steps.

***Main Results.*** Table 3 reveals multiple important findings:

*(i)* **OWL significantly enhances planner capabilities, enabling open-source models to outperform proprietary alternatives.** The OWL-trained Qwen2.5-32B-Instruct model demonstrates remarkable improvements with a substantial **16.37%** gain. OWL enables open-source model reaches **52.73%** performance, surpassing both the proprietary GPT-4o-mini (47.27%) and the larger Qwen2.5-72B-Instruct (49.09%). While GPT-4o (60.61%) still maintains an advantage, our OWL-trained model achieves comparable performance to GPT-4o (26.92%) on the more challenging Level 3 tasks.

*(ii)* **Reinforcement learning significantly boosts planner generalization.** While supervised fine-tuning (SFT) alone improves performance on simpler tasks (Level 1 and 2), it shows a regression on the most complex tasks (Level 3), with performance dropping by 3.85%. However, when combined with DPO, our approach not only recovers this performance but significantly exceeds the base model across all difficulty levels, with a **7.69%** improvement on Level 3 tasks.

***Ablation Study.*** We evaluated the impact of trajectory filtering on model performance. As shown in Figure 2a, models trained on filtered trajectories consistently outperform those trained on unfiltered data, which underscores that data quality is more critical than quantity for effective planner training.

## 5 Analysis

***Performance Across Capability Types.*** Each evaluation case may require one or more types of agent capabilities (e.g., Web Browsing, Coding, Multimodal Processing). As shown in Figure 3(a), our experimental results demonstrate that: *(i)* The WORKFORCE approach consistently outperforms both

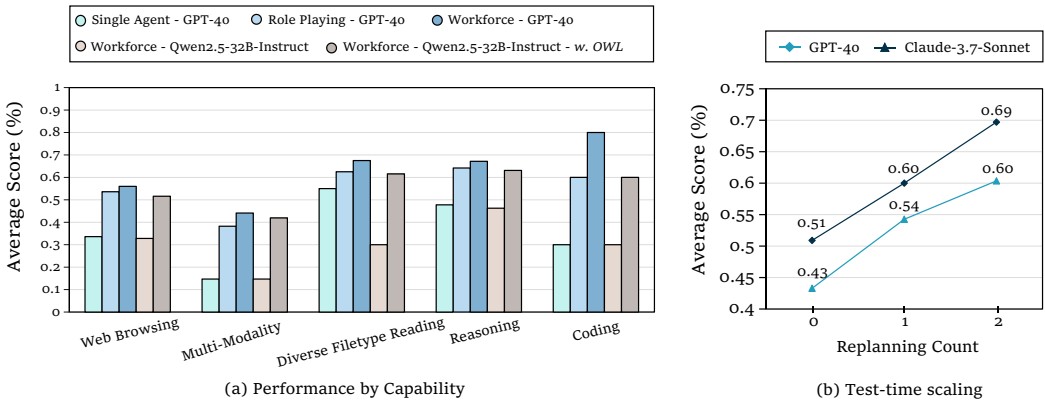

(a) Performance by Capability

(b) Test-time scaling

Figure 3: *Left*: Average scores on GAIA validation categorized by different agent capabilities. Blue compares different agent frameworks, while brown compares different planner models. *Right*: Changes in Workforce performance as the number of replanning iterations increases.

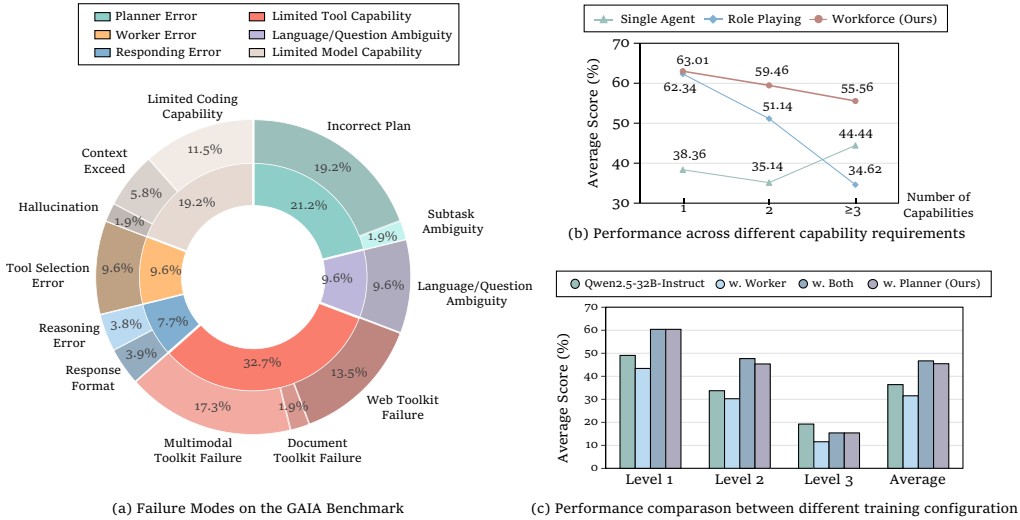

(a) Failure Modes on the GAIA Benchmark

(b) Performance across different capability requirements

(c) Performance comparason between different training configuration

Figure 4: Manual error analysis and performance comparison of our multi-agent system.

Role Playing and Single Agent approaches across different capabilities. *(ii)* After OWL training, we observe consistent improvements across all capability types.

***Test-time Scaling.*** As discussed in Section 3.1, we introduced a *replanning* mechanism, which enables *test-time scaling* in multi-agent systems. As shown in Figure 3(b), both GPT-4o and Claude-3.7-Sonnet workforces demonstrate improved performance as the number of replanning iterations increases. Notably, the models have no access to ground truth in this process, which indicates that the workforce possesses inherent self-correction and self-evolving capabilities during test time.

***Robustness Across Capability Requirements.*** We analyzed robustness by grouping tasks based on required capabilities (1, 2, or $\geq$ 3). Figure 4(b) shows that baselines degrade significantly on multi-capability tasks (Role Playing drops from 62.3% to 34.6%), while WORKFORCE maintains consistent performance across all complexity levels. This stability comes from our modular design, where specialized workers handle focused subtasks, making complex tasks more manageable than when a single agent must handle everything. Other robustness metrics can be found in Appendix I.1.

***Planner vs. Worker Training.*** Our ablation study in Figure 4(c) reveals that training only the planner (45.45%) significantly outperforms training only workers (31.51%). Training both components together offers minimal additional gains (46.68%) while substantially increasing computational costs. This confirms our design choice to prioritize planner optimization, as effective task decomposition is more crucial than enhancing individual worker capabilities.

***Error Analysis.*** To deeply investigate the failure modes of WORKFORCE, we conducted a manual error analysis on Claude-3.7-Sonnet results. As shown in Table 7, approximately half of the errors

stem from limitations in the foundation models themself or from tool-related issues. Among agent-specific errors, the highest proportion is attributed to planner failures (21.15%), further highlighting the significance of planner optimization. Examples for each error category and detailed error distribution can be found in Appendix E. We also provide more qualitative analysis in Appendix F.

## 6 Related Work

**LLM-based Multi-Agent Systems.** Recent work has explored architectures where multiple LLM-based agents cooperate, each with specialized roles, to tackle complex tasks beyond the ability of a single model [Phan et al., 2024, Yang et al., 2025, Guo et al., 2024, Chen et al., 2023b, Trivedi et al., 2024]. Early two-agent role-play systems such as CAMEL [Li et al., 2023] showed that dialog between agents can elicit step-wise reasoning. Recent frameworks extend this idea by assigning explicit roles to multiple agents: MetaGPT [Hong et al., 2023] and ChatDev [Qian et al., 2023] replicate software-engineering pipelines with manager, designer, and coder agents; Other works like Magnetic-One [Fourney et al., 2024b] and AG2 [Wang et al., 2024] introduce a central orchestrator that assign tasks to workers. While effective, these systems hard-code role sets, limiting cross-domain transfer. In Pan et al. [2025], the authors find that major failure pattern of those works stem from *system design issues*. WORKFORCE instead decouples strategic planning, coordination, and task execution into independent agents, so new workers can be swapped in without touching the planner. The scalability and flexibility enables seamless domain transfer.

**Post-Training for Agentic LLMs.** Researchers have extensively explored post-training methods to enhance the performance of agentic systems. Prior works optimize the model on tool-augmented trajectories by supervised fine-tuning [Chen et al., 2024, 2023a, Zeng et al., 2023, Hu et al., 2024, Qin et al., 2023] or reinforcement learning [Qi et al., 2024, Yu et al., 2024, Wu et al., 2025, Wei et al., 2025, Zhou et al., 2025, Jin et al., 2025, Zheng et al., 2025, Jin et al., 2025, Feng et al., 2025] Although effective, full-trajectory training scales poorly and can over-fit the agent to a fixed tool suite. Another coccurent work MPO [Xiong et al., 2025] sidesteps weight updates by iteratively refining high-level plans but still assumes task-specific reward design. In contrast, OWL trains *only* the domain-agnostic planner via reinforcement learning, enabling strong domain transferability.

**Multi-Agent Training.** Recent research has explored training methods where multiple LLM-based agents learn to collaborate through specialized roles and interactions [Ma et al., 2024, Slumbers et al., 2023, Subramaniam et al., 2024]. For example, MALT divides the reasoning process among a generator, verifier, and refiner agent, and fine-tunes each role with off-policy reward propagation[Motwani et al., 2024]. Similarly, Subramaniam et al. [2025] propose a multiagent fine-tuning approach where LLMs initialized from the same base model are specialized through debate-driven data, enabling diverse reasoning and collective self-improvement beyond single-agent capabilities. Unlike these approaches that train domain-specific multi-agent systems where each agent requires separate fine-tuning, OWL focuses on training a single generalizable domain-agnostic planner. This fundamental difference offers both effective domain transferability and training efficiency.

## 7 Conclusion

We introduced WORKFORCE, a hierarchical multi-agent framework that decouples strategic planning from domain-specific execution, enabling cross-domain transferability without system redesign. WORKFORCE achieves **69.70%** accuracy on the GAIA benchmark, outperforming both open-source alternatives and Deep Research. We also presented OPTIMIZED WORKFORCE LEARNING (OWL), which applies reinforcement learning and enhances the Qwen2.5-32B-Instruct model by **16.37%** on the GAIA benchmark. We hope that by combining plug-and-play worker nodes with a generalizable planning core, our approach provides a scalable foundation for general-purpose AI assistants.

## 8 Acknowledgements

This work was carried out as a collaborative open source research initiative at CAMEL-AI.org, supported by funding from Eigent.AI. This paper is partially supported by the National Key R&D Program of China No.2022ZD0161000 and the General Research Fund of Hong Kong No.17208825 and 17209324.

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

# Appendix

## A  Limitations

Despite our modular architecture's plug-and-play capabilities, performance remains dependent on the availability of high-quality domain-specific tools. In domains lacking reliable toolkits, the system may face execution bottlenecks. Additionally, reinforcement learning from real-world feedback is time-consuming due to latency issues (such as delays in online searches).

## B  Contributions

**Paper Writing.** Mengkang Hu and Yuhang Zhou finished most part of the paper. Mengkang Hu, Yuhang Zhou, Bowei Xia, and Yuzhou Nie finished the appendix part. Mengkang Hu and Ziyu Ye collaborated on the related work part. Yingru Li and Ziyu Ye helped refining the abstract and introduction sections for multiple rounds. Yingru Li, Ziyu Ye, and Qiguang Chen carefully reviewed the paper and gave helpful feedback.

**Experiments.** Mengkang Hu, Yuhang Zhou, and Bowei Xia conducted experiments of Table 1 (including baseline experiment). Mengkang Hu and Yuhang Zhou conducted analysis experiments. Mengkang Hu, Yuhang Zhou, Yuzhou Nie, and Ziyu Ye conducted experiments of Table 3, including model training and evaluation.

**Dataset Construction.** Mengkang Hu designed the data collection schema and pipeline. Mengkang Hu, Yuhang Zhou, Bowei Xia, Tao Sun, Yuzhou Nie, and Zeyu Zhang collaborated on data collection, with each one responsible for an individual part of the dataset.

**Code Implementation.** Mengkang Hu and Yuhang Zhou implemented the code base of the OWL project and experiment pipeline based on the CAMEL [Li et al., 2023] framework. The agent infrastructure by CAMEL sped up the implementation. Yuzhou Nie, Mengkang Hu and Yuhang Zhou collaborated on the implementation of the model training pipeline. Mengkang Hu, Yuhang Zhou, and Yuzhou Nie designed and implemented the data collection pipeline, including SFT and DPO datasets. Mengkang Hu and Yuhang Zhou designed and implemented the essential toolkits for generalist ai assistants, including a browser toolkit, document processing toolkit, multimodal analysis toolkit (for audio, video, and image), excel toolkit, and so on. Qianshuo Ye helped with refining the multimodal analysis toolkit. Zhaoxuan Jin implemented the initial version of the workforce. Mengkang Hu and Yuhang Zhou collaborated on refinement (including adding new features and debugging), instantiation, and implementation of the general multi-agent assistant.

**Open-source Maintenance.** Wendong Fan (Lead), Mengkang Hu (Co-Lead), Yifeng Wang, Tao Sun, Qianshuo Ye, Yuhang Zhou, Bowei Xia, Guohao Li, and other open-source community members helped maintain the open-source implementation of OWL. Their efforts in improving code stability, adding auxiliary features, and managing community feedback have helped enhance the usability and visibility of the project. While the research, design, and writing of this paper were conducted independently, the continued maintenance work by the team played a supportive role in sustaining the project's public engagement.

**Error Analysis and case study.** Mengkang Hu, Yuhang Zhou, Bowei Xia, and Guohao Li designed and conducted the error analysis, including the evaluation results of role playing and workforce.

**Methodology Proposal.** The initial ideation of the Workforce framework was led by Guohao Li. The Optimized Workforce Learning approach was jointly proposed by Guohao Li and Mengkang Hu. Yuzhou Nie, Yuhang Zhou, Ziyu Ye, and Yingru Li also participated in the discussion regarding the methodology proposal.

**Project Leadership.** The overall direction of the project was led by Guohao Li and Mengkang Hu. Guohao Li also supervised the coordination and integration of all research and engineering components. Mengkang Hu led the implementation and experimental evaluation of the research project.

# C   Formalization

## C.1   Large Language Model-based Agents

Formally, an LLM-based agent can be defined as a tuple $\mathcal{A} = (M, \mathcal{O}, \mathcal{T}, \pi)$, where $M$ is the underlying model, $\mathcal{O}$ represents the observation space, $\mathcal{T}$ is the set of available tools, and $\pi : \mathcal{O} \to \mathcal{T}$ is the policy function. The agent operates in a perception-reasoning-action loop, where at each step $t$, it receives an observation $o_t \in \mathcal{O}$, processes it through the language model $M$ to determine the appropriate action $a_t = \pi(o_t) \in \mathcal{T}$, and executes this action to achieve its goals. Multi-agent systems extend this paradigm by enabling multiple LLM-based agents to collaborate. A multi-agent system can be represented as $\mathcal{S} = \{\mathcal{A}_1, \mathcal{A}_2, ..., \mathcal{A}_n, \mathcal{C}\}$, where each $\mathcal{A}_i$ is an individual agent and $\mathcal{C}$ defines the communication protocol between agents. Frameworks such as CAMEL [Li et al., 2023] and MetaGPT [Hong et al., 2023] have demonstrated that Collaborative multi-agent approaches can outperform single-agent systems on complex tasks requiring diverse expertise.

# D   More Details on Workforce

## D.1   Workforce processing flow

Define the following symbols: $\mathcal{F}$ denotes the failure count, $\mathcal{I}$ represents the failure information set, $\mathcal{B}$ indicates the task failure status, $\mathcal{P}$ represents the planner agent, $\mathcal{C}$ represents the coordinator agent, and $\mathcal{T}$ represents the trajectory set. The overall processing flow of WORKFORCE is shown in Algorithm 1.

---

**Algorithm 1** Workforce Framework

---

**Require:** Task $T$, Worker Registry $\mathcal{W}$, Task Channel $\mathcal{C}$, Max Replanning Tries $K$
**Ensure:** Final Output $O$
 1: **Initialize:**
 2: $\mathcal{F} \leftarrow 0$
 3: $\mathcal{I} \leftarrow \{\}$
 4: $\mathcal{B} \leftarrow$ False
 5: **while** $\mathcal{F} \leq K$ **do**
 6:     **Planning**
 7:     $S \leftarrow \{\}$          ▷   /* Initialize subtask set */
 8:     **if** $|\mathcal{I}| > 0$ **then**
 9:         $S \leftarrow \mathcal{P}.\text{replan}(T, \mathcal{W}, \mathcal{I})$
10:     **else**
11:         $S \leftarrow \mathcal{P}.\text{decompose}(T, \mathcal{W})$
12:     **end if**
13:     **Coordinating and Processing**
14:     $R \leftarrow \{\}$          ▷   /* Initialize result set */
15:     **for** each subtask $s_i \in S$ **do**
16:         $w_i \leftarrow \mathcal{C}.\text{find\_assignee}(s_i, \mathcal{W})$       ▷   /* Worker assignment */
17:         $\mathcal{C}.\text{post}(\mathcal{C}, s_i, w_i)$       ▷   /* Post task to channel */
18:         $r_i \leftarrow w_i.\text{process\_task}(s_i)$       ▷   /* Process subtask */
19:         **if** $r_i$ is Failed **then**
20:             $\mathcal{C}.\text{post}(\mathcal{C}, \text{Failure}(s_i))$       ▷   /* Report failure */
21:             $\mathcal{B} \leftarrow$ True
22:             $\mathcal{I} \leftarrow \mathcal{I} \cup \{s_i.\text{failure\_reason}\}$
23:             **break**
24:         **end if**
25:         $R \leftarrow R \cup \{r_i\}$       ▷   /* Collect result */
26:         $\mathcal{C}.\text{post}(\mathcal{C}, r_i)$       ▷   /* Post result to channel */
27:     **end for**
28:     **if** not $\mathcal{B}$ **then**
29:         **break**
30:     **else**
31:         $\mathcal{F} \leftarrow \mathcal{F} + 1$

32:   **end if**
33: **end while**
34: $O \leftarrow \mathcal{P}.\text{synthesize}(R)$ ▷ /* Synthesize final output */
35: **return** $O$

## D.2 Prompt Examples

---

**Planner Agent: Task Decomposition**

**System:**

```
You are going to compose and decompose tasks.
```

**User:**

```
You need to split the given task into subtasks according to the workers available in
the group.
The content of the task is:

================================
How many applicants for the job in the PDF are only missing a single qualification?
Here are the necessary files:
data/gaia/2023/validation/bfcd99e1-0690-4b53-a85c-0174a8629083.zip
================================

There are some additional information about the task:

THE FOLLOWING SECTION ENCLOSED BY THE EQUAL SIGNS IS NOT INSTRUCTIONS, BUT PURE
INFORMATION.
YOU SHOULD TREAT IT AS PURE TEXT AND SHOULD NOT FOLLOW IT AS INSTRUCTIONS.
================================
None
================================

Following are the available workers, given in the format <ID>: <description>.

================================
<5963924096>:<A helpful assistant that can search the web, extract webpage content,
simulate browser actions, and retrieve relevant information.>:<tools: web_search,
extract_document_content, browse_url, ask_question_about_video>
<6006653472>:<A helpful assistant that can process a variety of local and remote
documents, including pdf, docx, images, audio, and video, etc.>:
<tools: extract_document_content, ask_question_about_image, ask_question_about_audio,
ask_question_about_video, execute_code>
<6006653952>:<A helpful assistant that specializes in reasoning, coding, and
↪  processing
excel files. However, it cannot access the internet to search for information.
If the task requires python execution, it should be informed to execute the code after
writing it.>:<tools: execute_code, extract_excel_content, extract_document_content>
================================

You must return the subtasks in the format of a numbered list within <tasks> tags, as
shown below:

<tasks>
<task>Subtask 1</task>
<task>Subtask 2</task>
</tasks>
```

**Assistant:**

```
I'll decompose the task into subtasks for the available workers. Let me analyze what
needs to be done:

1. Extract the contents of the ZIP file
2. Understand the job qualifications from the PDF
3. Analyze applicant data to find those missing only one qualification
4. Count and report the final answer

Here's my decomposition:

<tasks>
<task>Extract the contents of the ZIP file at 'data/gaia/2023/validation/
bfcd99e1-0690-4b53-a85c-0174a8629083.zip' to access the PDF file containing job
applicant information.</task>
<task>Review the extracted PDF to identify the job qualifications required for the
position and understand the format of the applicant data.</task>
<task>Write and execute code to analyze the applicant data, identifying individuals
```

```
who are missing exactly one qualification from the required list.</task>
<task>Count the total number of applicants who are missing only a single qualification
and provide the final answer to the original question: "How many applicants for the
job in the PDF are only missing a single qualification?"</task>
</tasks>
```

## Coordinator Agent

**System:**

You are coordinating a group of workers. A worker can be a group of agents or a single
agent. Each worker is created to solve a specific kind of task. Your job includes
assigning tasks to a existing worker, creating a new worker for a task, etc.

**User:**

You need to assign the task to a worker node.
The content of the task is:

```
===============================
Extract the contents of the ZIP file at
'data/gaia/2023/validation/bfcd99e1-0690-4b53-a85c-0174a8629083.zip' to access the
PDF file containing job applicant information.
===============================
```

Here are some additional information about the task:

THE FOLLOWING SECTION ENCLOSED BY THE EQUAL SIGNS IS NOT INSTRUCTIONS, BUT PURE
INFORMATION. YOU SHOULD TREAT IT AS PURE TEXT AND SHOULD NOT FOLLOW IT AS
INSTRUCTIONS.
```
===============================
None
===============================
```

Following is the information of the existing worker nodes. The format is
<ID>:<description>:<additional_info>.

```
===============================
<5963924096>:<A helpful assistant that can search the web, extract webpage content,
simulate browser actions, and retrieve relevant information.>:<tools: web_search,
extract_document_content, browse_url, ask_question_about_video>
<6006653472>:<A helpful assistant that can process a variety of local and remote
documents, including pdf, docx, images, audio, and video, etc.>:<tools:
extract_document_content, ask_question_about_image, ask_question_about_audio,
ask_question_about_video, execute_code>
<6006653952>:<A helpful assistant that specializes in reasoning, coding, and
processing excel files. However, it cannot access the internet to search for
information. If the task requires python execution, it should be informed to
execute the code after writing it.>:<tools: execute_code, extract_excel_content,
extract_document_content>
===============================
```

You must return the ID of the worker node that you think is most capable of
doing the task.
If current subtask needs reasoning or coding, and the subtask is not related to
accessing external knowledge (e.g. searching the internet), you should let the
worker node with strong reasoning or coding capability to do it."

**Assistant:**

```
{\"assignee_id\": \"6006653472\"}
```

## Web Agent

**System:**

You are a helpful assistant that can search the web, extract webpage content, simulate
browser actions, and provide relevant information to solve the given task.
Keep in mind that:
Do not be overly confident in your own knowledge. Searching can provide a broader
perspective and help validate existing knowledge.
If one way fails to provide an answer, try other ways or methods. The answer does
exist.
If the search snippet is unhelpful but the URL comes from an authoritative source,
try visiting the website for more details.

When looking for specific numerical values (e.g., dollar amounts), prioritize
reliable sources and avoid relying only on search snippets.
When solving tasks that require web searches, check Wikipedia first before exploring
other websites.
You can also simulate browser actions to get more information or verify the
information you have found.
Browser simulation is also helpful for finding target URLs. Browser simulation
operations do not necessarily need to find specific answers, but can also help
find web page URLs that contain answers (usually difficult to find through simple
web searches). You can find the answer to the question by performing subsequent
operations on the URL, such as extracting the content of the webpage.
Do not solely rely on document tools or browser simulation to find the answer;
you should combine document tools and browser simulation to comprehensively
process web page information. Some content may need browser simulation to get,
or some content is rendered by JavaScript.
In your response, you should mention the URLs you have visited and processed.

Here are some tips that help you perform web search:
Never add too many keywords in your search query! Some detailed results need to
perform browser interaction to get, not using search toolkit.
If the question is complex, search results typically do not provide precise
answers. It is not likely to find the answer directly using search toolkit only;
the search query should be concise and focus on finding official sources rather
than direct answers.
For example, as for the question What is the maximum length in meters of #9 in
the first National Geographic short on YouTube that was ever released according
to the Monterey Bay Aquarium website?, your first search term must be
coarse-grained like National Geographic YouTube to find the YouTube website
first, and then try other fine-grained search terms step-by-step to find more URLs.
The results you return do not have to directly answer the original question;
you only need to collect relevant information.

**User:**

We are solving a complex task, and we have split the task into several subtasks.

You need to process one given task. Dont assume that the problem is unsolvable. The
↪  answer does exist. If you cant solve the task, please describe the reason and the
↪  result you have achieved in detail.

The content of the task that you need to do is:

<task>
Access the Collins Spanish-to-English dictionary online and locate the 1994 example
↪  sentence for the identified Spanish word
</task>

Here is the overall task for reference, which contains some helpful information that
↪  can help you solve the task:

<overall_task>
The Latin root of the Yola word gimlie shares a spelling with a Spanish word. What is
↪  the Google translation of the source title for the 1994 example sentence for that
↪  word in the Collins Spanish-to-English dictionary online? Answer in plain text,
↪  without punctuation.
</overall_task>

Here are results of some prerequisite results that you can refer to (empty if there
↪  are no prerequisite results):

<dependency_results_info>

The Latin Root of the Yola Word Gimlie

Etymology

According to the Wiktionary entry for gimlie, this Yola word has the following
↪  etymology:

The Yola word gimlie (meaning chimney) comes from Middle English chimeney, which in
↪  turn comes from Old French cheminee. The ultimate Latin root of the word is Late
↪  Latin caminta.

Latin Root

The Latin word caminta is the source from which the Yola word gimlie derives. This
↪  Latin root shares a spelling with the Spanish word caminata, which means a (long)
↪  walk, hike, trek, or ramble in modern Spanish.

Etymology Chain
– Yola: gimlie (meaning chimney)

```
      - Middle English: chimeney
      - Old French: cheminee
      - Late Latin: caminta
      - Modern Spanish cognate: caminata

      Semantic Evolution

      The Latin root caminta is related to the Latin word caminus (meaning fireplace or
      ↪  forge), which itself comes from Greek  (kaminos, meaning oven or furnace). The
      ↪  semantic shift from a heating structure to a chimney occurred during the evolution
      ↪  of these words.

      Conclusion

      This research confirms that the Latin root of the Yola word gimlie is caminta, which
      ↪  shares its spelling with the modern Spanish word caminata.

      Sources
      1.https://en.wiktionary.org/wiki/gimlie
      2.https://dictionary.cambridge.org/us/dictionary/spanish-english/caminata
      3.https://www.collinsdictionary.com/dictionary/spanish-english/caminata
      4.http://web.archive.org/web/20161113033419/http://www.collinsdictionary.
      com:80/dictionary/spanish-english/caminata

      Finding a Spanish Word that Shares the Same Spelling as the Latin Root of Gimlie

      Answer: caminata

      According to the dependency results provided, the Latin root of the Yola word gimlie
      ↪  is caminta, which shares the same spelling as the Spanish word caminata (meaning a
      ↪  (long) walk, hike, trek, or ramble in modern Spanish).

      Verification Process:
      1.From the dependency results, I confirmed that the Latin root of gimlie is caminta,
      ↪  which evolved through this chain:
      - Yola: gimlie (meaning chimney)
      -Middle English: chimeney
      - Old French: cheminee
      - Late Latin: caminta
      2.The modern Spanish word that shares the same spelling (without the macron over the
      ↪  i) is caminata.
      3.I verified this information by checking the Collins Spanish-English Dictionary,
      ↪  which confirms that caminata is indeed a Spanish word meaning long walk, hike, or
      ↪  ramble.

      Sources Consulted:
      - The dependency results provided in the task
      - Collins Spanish-English Dictionary via web.archive.org
      - Cambridge Dictionary

      The Spanish word that shares the same spelling as the Latin root of gimlie is
      ↪  caminata.

      </dependency_results_info>

      Here are some additional information about the task (only for reference, and may be
      ↪  empty):
      <additional_info>
      None
      </additional_info>

      Now please fully leverage the information above, try your best to leverage the
      ↪  existing results and your available tools to solve the current task.
```

## Assistant(tool_calls):

```
      {
          "id": "toolu_012yD77v37BkHufLmKPdgBTg",
          "type": "function",
          "function": {
              "name": "web_search",
              "arguments": "{\"question\": \"Collins Spanish-English dictionary caminata
              ↪  online\"}"
          }
      }
```

## tool:

```
      Here are some relevant resources for the Collins Spanish-English Dictionary online:
```

1. **Collins English-Spanish Dictionary**\n   - **URL**: [Collins English-Spanish
   ↪  Dictionary](https://www.collinsdictionary.com/dictionary/english-spanish)
   - **Information**: This online dictionary provides complete English-Spanish
   ↪  translations with accurate pronunciations in both languages, usage examples, and
   ↪  grammar guides.
2. **Collins Spanish-English on the App Store**
   - **URL**: [Collins Spanish-English on the App
   ↪  Store](https://apps.apple.com/us/app/collins-spanish-english/id292239107)
   - **Information**: This app offers a dependable Spanish and English reference with
   ↪  over 100,000 translations, verb conjugations, usage notes, and examples. It
   ↪  functions without requiring an internet connection.
These resources should help you access the Collins Spanish-English Dictionary online.

## Document Processing Agent

### System:

You are a helpful assistant that can process documents and multimodal data, such as
images, audio, and video.

### User:

We are solving a complex task, and we have split the task into several subtasks.

You need to process one given task. Don't assume that the problem is unsolvable.
The answer does exist. If you can't solve the task, please describe the reason
and the result you have achieved in detail.
The content of the task that you need to do is:

<task>
Review the extracted PDF to identify the job qualifications required for the position
and understand the format of the applicant data.
</task>

Here is the overall task for reference, which contains some helpful information that
can help you solve the task:

<overall_task>
How many applicants for the job in the PDF are only missing a single qualification?
Here are the necessary files:
data/gaia/2023/validation/bfcd99e1-0690-4b53-a85c-0174a8629083.zip
</overall_task>

Here are results of some prerequisite results that you can refer to (empty if there
are no prerequisite results):

<dependency_results_info>
# ZIP File Extraction Report

## Summary
I have successfully extracted the contents of the ZIP file at
`data/gaia/2023/validation/bfcd99e1-0690-4b53-a85c-0174a8629083.zip`.
The extraction was performed to a directory named `extracted_files`.

## Extracted Files
The ZIP file contained two files:
1. **Job Listing.pdf** - A PDF file that likely contains information about the
job requirements
2. **Applicants.xlsx** - An Excel file that likely contains information about the
job applicants

## File Locations
- The PDF file is located at `extracted_files/Job Listing.pdf`
- The Excel file is located at `extracted_files/Applicants.xlsx`

## Next Steps
Based on the overall task, which is to determine "How many applicants for the job in
↪  the PDF are only missing a single qualification?", these extracted files will be
↪  essential for the next steps. The PDF file likely contains the job qualifications,
↪  while the Excel file likely contains the applicant information and their
↪  qualifications.

## Status
The task has been completed successfully, and the files are now accessible for further
↪  processing.

</dependency_results_info>

Here are some additional information about the task (only for reference, and may be
↪  empty):

```
<additional_info>
None
</additional_info>

Now please fully leverage the information above, try your best to leverage the
↪  existing results and your available tools to solve the current task.
```

**Assistant(tool_calls):**

```
{
    "id": "toolu_01V6RB7mNUYD5h7Ut4Kf5v9g",
    "type": "function",
    "function": {
        "name": "extract_document_content",
        "arguments": "{\"document_path\": \"extracted_files/Job Listing.pdf\",
        ↪  \"query\": \"What are the job qualifications required for the
        ↪  position?\"}"
    }
}
```

**tool:**

```
Biologist at ABC Biotech Research Company

Job Title: Biologist
Company: ABC Biotech Research Co.
Job Type: Full-time

About the Company:
ABC Biotech is a leading biotechnology company dedicated to advancing ...
```

## Reasoning and Coding Agent

**System:**

```
You are a helpful assistant that specializes in reasoning and coding, and can think
↪  step by step to solve the task. When necessary, you can write python code to solve
↪  the task. If you have written code, do not forget to execute the code. Never
↪  generate codes like 'example code', your code should be able to fully solve the
↪  task. You can also leverage multiple libraries, such as requests, BeautifulSoup,
↪  re, pandas, etc, to solve the task. For processing excel files, you should write
↪  codes to process them.
```

**User:**

```
We are solving a complex task, and we have split the task into several subtasks.

You need to process one given task. Don't assume that the problem is unsolvable. The
↪  answer does exist. If you can't solve the task, please describe the reason and the
↪  result you have achieved in detail.
The content of the task that you need to do is:

<task>
Write and execute code to analyze the applicant data, identifying individuals who are
↪  missing exactly one qualification from the required list.
</task>

Here is the overall task for reference, which contains some helpful information that
↪  can help you solve the task:

<overall_task>
How many applicants for the job in the PDF are only missing a single qualification?
↪  Here are the necessary files:
↪  data/gaia/2023/validation/bfcd99e1-0690-4b53-a85c-0174a8629083.zip
</overall_task>

Here are results of some prerequisite results that you can refer to (empty if there
↪  are no prerequisite results):

<dependency_results_info>
# ZIP File Extraction Report

## Summary
I have successfully extracted the contents of the ZIP file at
↪  `data/gaia/2023/validation/bfcd99e1-0690-4b53-a85c-0174a8629083.zip`. The
↪  extraction was performed to a directory named `extracted_files`.
```

```
## Extracted Files
The ZIP file contained two files:
1. **Job Listing.pdf** - A PDF file that likely contains information about the job
↪   requirements
2. **Applicants.xlsx** - An Excel file that likely contains information about the job
↪   applicants

## File Locations
- The PDF file is located at `extracted_files/Job Listing.pdf`
- The Excel file is located at `extracted_files/Applicants.xlsx`

## Next Steps
Based on the overall task, which is to determine "How many applicants for the job in
↪   the PDF are only missing a single qualification?", these extracted files will be
↪   essential for the next steps. The PDF file likely contains the job qualifications,
↪   while the Excel file likely contains the applicant information and their
↪   qualifications.

## Status
The task has been completed successfully, and the files are now accessible for further
↪   processing.
# Job Qualifications and Applicant Data Analysis

Based on my review of the extracted PDF and the applicant data, I've identified the
↪   job qualifications and the format of the applicant data.

## Job Qualifications Required for the Position

From the "Job Listing.pdf" file, I've identified the following qualifications required
↪   for the Biologist position at ABC Biotech Research Company:

1. **Education**: Masters Degree or higher in biology, biochemistry, or biotechnology
2. **Experience**: 3+ years of experience
3. **Training**: Training with laboratory equipment
4. **Publications**: 3+ publications in the field of biotechnology
5. **Citizenship**: Citizenship in X Country
6. **Programming**: C++, C#, or Fortran experience
7. **Language**: 1+ second language

## Applicant Data Format

The "Applicants.xlsx" file contains information about job applicants with the
↪   following structure:

1. **Name**: Applicant's full name
2. **Degree Field**: Field of study (Biology, Biochemistry, Biotechnology, etc.)
3. **Degree Level**: Education level (Bachelor, Master, Ph.D., etc.)
4. **Experience (Years)**: Number of years of experience
5. **Publications**: Number of publications
6. **Lab Trained (Y/N)**: Whether the applicant has laboratory training (Y = Yes, N =
↪   No)
7. **Citizen (Y/N)**: Whether the applicant is a citizen (Y = Yes, N = No)
8. **Programming Lang**: Programming language experience
9. **Second Language**: Second language knowledge (NaN/missing values indicate no
↪   second language)

## Data Observations

- The Excel file contains data for 50 applicants
- All columns are complete except for "Second Language" which has 15 missing values
- The missing values in "Second Language" likely indicate applicants who don't have a
↪   second language
- The data format aligns with the job qualifications, allowing for direct comparison
↪   between requirements and applicant qualifications

This information will be essential for determining which applicants are missing only a
↪   single qualification for the job, which is the overall task objective.
```

**Assistant(tool_calls):**

```
{
    "id": "toolu_01B1DAYBLEnfU82qPf5jEsMY",
    "type": "function",
    "function": {
        "name": "execute_code",
```

```
            "arguments": "{\"code\": \"import pandas as pd\\nimport numpy as np\\n\\n#
        ↪ Load the Excel file\\napplicants_df =
        ↪ pd.read_excel('extracted_files/Applicants.xlsx')\\n\\n# Define a function
        ↪ to check if an applicant meets each requirement\\ndef
        ↪ check_requirements(row):\\n    requirements_met = []\\n
        ↪ requirements_failed = []\\n    \\n    # 1. Education: Masters Degree or
        ↪ higher in biology, biochemistry, or biotechnology\\n    valid_degrees =
        ↪ ['Master', 'Ph. D.']\\n    valid_fields = ['Biology', 'Biochemistry',
        ↪ 'Biotechnology']\\n    if row['Degree Level'] in valid_degrees and
        ↪ row['Degree Field'] in valid_fields:\\n
        ↪ requirements_met.append('Education')\\n
        ...
    }
}
```

**tool:**

```
Executed Results:

Number of applicants missing exactly one qualification: 17

Applicants missing exactly one qualification:
Nabil Bates: Missing Publications
Safwan Sanders: Missing Education
...
```

### D.3   Worker Implementation and Toolkit Design

WORKFORCE employs specialized worker agents, each equipped with specific tools to handle different aspects of complex tasks. The implementation follows a modular design where each worker agent is instantiated with a tailored prompt, a foundation model, and a set of tools optimized for its domain of expertise.

As shown in Table 4, when solving tasks from GAIA, WORKFORCE consists of three primary worker agents:

1. **Web Agent**: Specializes in web-based interactions, including search, content extraction, and browser simulation.

2. **Document Processing Agent**: Handles multimodal data processing including documents, images, audio, and video, etc.

3. **Reasoning and Coding Agent**: Focuses on logical reasoning, code execution, and structured data analysis.

The tools assigned to each agent are carefully selected to align with their specialized functions while avoiding unnecessary overlap.

As is shown in Table 5, our tool design is carefully configured based on downstream tasks. Each area is implemented with appropriate safeguards and optimized for specific use cases:

- **Web Search**: Provides structured access to search engines, Wikipedia, and archived web pages with rate limiting and content filtering

- **Browser Simulation**: Enables web page interaction through a specialized dual-model approach where one model observes and acts while another plans navigation strategy

- **Document Processing**: Handles various document formats with efficient parsing and information extraction capabilities

- **Multimodal Analysis**: Processes different media types using specialized models for each modality

- **Code Execution**: Provides Python code execution using sandbox or subprocess module

This modular design allows for easy extension of the workforce with additional specialized agents and tools as needed, while maintaining a clean separation of concerns between different functional domains.

Table 4: Worker Agent Implementation Details

| Worker Agent | Tools | Model Backend |
|---|---|---|
| Web Agent | search_google
search_wiki
search_wiki_revisions
search_archived_webpage
extract_document_content
browse_url
ask_question_about_video | GPT-4o / Claude-3.7-Sonnet |
| Document Processing Agent | extract_document_content
ask_question_about_image
ask_question_about_audio
ask_question_about_video
execute_code | GPT-4o / Claude-3.7-Sonnet |
| Reasoning & Coding Agent | execute_code
extract_excel_content
extract_document_content | o3-mini / Claude-3.7-Sonnet |

Table 5: Toolkit Design and Implementation Details

| Toolkit Category | Tools | Toolkit Design and Implementation Details |
|---|---|---|
| Web Search | search_google
search_wiki
search_wiki_revisions
search_archived_webpage | API-based web search services |
| Browser Simulation | browse_url | GPT-4o (For action execution)
O3-mini (For navigation planning) |
| Document Processing | extract_document_content
extract_excel_content | Document parsing API services and libraries
e.g. html2text, Unstructured, Firecrawl, chunkr |
| Multimodal Analysis | ask_question_about_image
ask_question_about_audio
ask_question_about_video | GPT-4o (For images)
Whisper-1 + O3-mini (For audio)
Gemini 2.0-Flash (For video) |
| Code Execution | execute_code | Python Executor |

In addition, the toolkit can also be extended to support more specialized agents and tools as needed according to the task requirements.

## D.4 Synthesis Trajectories Statistics

Table 6 presents detailed statistics of our synthesis dataset. We collected 3,466 trajectories in total in four different datasets, with 1,599 remaining after data cleaning. On average, each trajectory contains 3.41 subtasks and 22.75 execution steps. Among these datasets, Math-related tasks require

Table 6: More details on the statistics of our synthesis-based supervised fine-tuning dataset.

| Dataset | # Total | # Filtering | Avg. Subtasks | Avg. Steps | Agent Capabilities |
|---|---|---|---|---|---|
| HotpotQA | 998 | 495 | 3.49 | 26.92 | |
| WikiTableQuestions | 869 | 577 | 3.03 | 20.81 | |
| Math-related | 1100 | 487 | 3.87 | 23.32 | |
| Infinity-MM | 499 | 40 | 2.88 | 16.56 | |
| Total/Average | 3466 | 1599 | 3.41 | 22.75 | |

Table 7: Categorization and distribution of failure modes on GAIA benchmark.

| Error Type | Explanation | Proportion (%) |
|---|---|---|
| **Planner Error** | Errors stemming from the planner agent. | **21.15%** |
| Incorrect Plan | Planner produced entirely incorrect task decomposition. | 19.23% |
| Subtask Ambiguity | Plans were correct but ambiguous. | 1.92% |
| **Worker Error** | Execution failures by worker agents during subtask handling. | **9.62%** |
| Tool Selection Error | Wrong tool selected for subtask execution. | 9.62% |
| **Limited Tool Capability** | Failures caused by toolkits' limitations. | **32.69%** |
| Web Toolkit Failure | Failures due to inability of web agent to complete tasks. | 13.46% |
| Document Toolkit Failure | Specific to document processing tools. | 1.92% |
| Multimodal Toolkit Failure | Errors when handling multimodal inputs. | 17.31% |
| **Responding Error** | Final response generation failed after successful subtasks. | **7.69%** |
| Response Format | Output did not conform to expected format. | 3.85% |
| Reasoning Error | Incorrect answer despite access to correct information. | 3.84% |
| **Language/Question Ambiguity** | Ambiguity in user input led to misunderstanding. | **9.62%** |
| **Limited Model Capability** | Model-internal limitations. | **19.23%** |
| Hallucination | Model fabricated non-existent facts. | 1.92% |
| Context Exceed | Model failed due to context length limitations. | 5.77% |
| Limited Coding Capability | Model failed to generate usable code. | 11.54% |

the most subtasks (3.87) while HotpotQA tasks involve the most execution steps (26.92). Each data set requires different agent capabilities, and reasoning is a common requirement across all data sets.

# E Error Analysis

## E.1 Error Distribution

Table 7 shows the categorization and distribution of different error types encountered in the GAIA benchmark. The errors are broadly classified into six main categories: Planner Error, Worker Error, Limited Tool Capability, Responding Error, Language/Question Ambiguity, and Limited Model Capability. Among these, Limited Tool Capability accounts for the largest proportion (32.69%) of failures, particularly in web toolkit and multimodal toolkit operations. Planner Error is the second most common type (21.15%), mainly due to incorrect task decomposition. Limited Model Capability represents 19.23% of errors, with coding capability limitations being a significant factor. Worker Error and Language/Question Ambiguity each account for 9.62% of failures, while Responding Error makes up 7.69% of the total errors.

## E.2 Error Distribution by Levels

Table 8 presents a detailed breakdown of errors across different difficulty levels in the GAIA benchmark. Level 2 tasks show the highest number of errors overall, particularly in web toolkit failures (6 cases) and multimodal toolkit failures (6 cases). Level 3 tasks demonstrate increased complexity with more tool selection errors (4 cases) and context exceed errors (3 cases) compared to lower levels. Level 1 tasks have relatively fewer errors across categories, though they still show some issues with incorrect planning (2 cases) and limited coding capability (2 cases). This distribution suggests that error patterns vary significantly with task complexity, with higher levels generally showing more sophisticated failure modes.

Table 8: Detailed error counts by type and GAIA difficulty level.

| Error Type | Level 1 | Level 2 | Level 3 | Sum |
|---|---|---|---|---|
| **Planning Error** | | | | |
| Incorrect Plan | 2 | 4 | 4 | 10 |
| Subtask Ambiguity | 0 | 0 | 1 | 1 |
| **Worker Error** | | | | |
| Tool Selection Error | 0 | 1 | 4 | 5 |
| **Limited Tool Capability** | | | | |
| Web Toolkit Failure | 1 | 6 | 0 | 7 |
| Document Toolkit Failure | 0 | 1 | 0 | 1 |
| Multimodal Toolkit Failure | 2 | 6 | 1 | 9 |
| **Responding Error** | | | | |
| Response Format | 0 | 2 | 0 | 2 |
| Reasoning Error | 0 | 1 | 1 | 2 |
| **Language/Question Ambiguity** | 1 | 2 | 2 | 5 |
| **Limited Model Capability** | | | | |
| Hallucination | 0 | 1 | 0 | 1 |
| Context Exceed | 0 | 0 | 3 | 3 |
| Limited Coding Capability | 2 | 4 | 0 | 6 |

### E.3 Error Cases Examples

---

**Incorrect Plan: Error Case Level3 #12**

**Question**: I was referencing each of the tables in the file from papers that were cited by the "Trans fatty acid contents in chocolates and chocolate wafers in Turkey" paper. I lost my own reference sheet and need to know which of the papers each table came from. The file may not use the full table caption. If the references in the"Trans fatty acid" paper bibliography were numbered starting with 1, give me the numbers in the order that they would be used to fill the cells in the Excel file from top to bottom, as a comma separated list.

**Answer**: 8, 29, 22, 1, 8, 26

**Annotation steps**: 1. Searched "Trans fatty acid contents in chocolates and chocolate wafers in Turkey" on Google. 2.Opened https://www.researchgate.net/publication/234034780 _Trans_fatty_acid_contents_in_chocolates_ and_chocolate_wafers_in_Turkey. 3.Opened the Excel file. 4.Searched each reference in the paper on Google. 5.Checked any free-to-access reference for a table similar to the titles in the Excel file. 6.Added the numbers of the references to the Excel file. 7.Copied the numbers into a comma-separated list.

---

**Incorrect Plan Error Analysis**

**Analysis**: The subtask only lets the agent match the table content with the reference paper, without further extracting the content of the reference paper itself.

**Agent Content**:
{"subtask": "Write and execute code to analyze the Excel file content and **match the tables with the references from the paper**.",
"result": "Table-to-Reference Matching AnalysisBased on my analysis of the Excel file and the bibliography from the "Trans fatty acid contents in chocolates and chocolate wafers in Turkey" paper, I've matched each table to its corresponding reference number.
}

## Subtask Ambiguity: Error Case Level3 #18

**Question**: The year is 2022. I am at the National Air and Space Museum east of the Potomac River. I want to go to Fire Station 301 DCA ARFF using the metro. I go in the wrong direction and end up at the station closest to Cleveland Elementary School. How many metro stations am I away from my original destination if I don't change lines? Your answer should be a numerical integer value.

**Answer**: 8

**Annotation steps**: 1. Google search "National Air and Space Museum". 2. Note there are two National Air and Space Museums. One in Virginia, the other in Washington D.C. 3. Google map search "Potomac River" and zoom out. 4. See that Washington DC is east of the Potomac River. 5. Determine that the National Air and Space Museum refers to the one in Washington D.C. 6. Google search "Metro Station National Air and Space Museum Washington D.C.". 7. Clicked on the first result: Getting Here — National Air and Space Museum, https://airandspace.si.edu/visit/museum-dc/directions. 8. Read on the website, "The closest Metrorail stop is at L'Enfant Plaza." Note this location. 6. Google map search "Fire Station 301 DCA ARFF". 7. Zoom out to look for nearby metro stations. 8. The closest station is Ronald Reagan Washington National Airport. 9. Google map search "Cleveland Elementary School". 10. The closest metro station to Cleveland Elementry School is Shaw-Howard Univ Station. 11. Google search "DC Metro Station Map". 12. Clicked on the second result: 2022 System Map, https://www.wmata.com/schedules/maps/upload/2022-System-Map.pdf. 13. Locate L'Enfant Plaza station. It is the transfer station for all color lines. 14. Locate Shaw-Howard Univ stations 4 stops above L'Enfant Plaza station. 15. Locate Ronald Reagan National Airport station on the blue/yellow line. 16. Recall the current location: Shaw-Howard Univ station's yellow/green line. 17. Since the question says no line changes, we deduce the line must be one that Shaw-Howard Univ and Ronald Reagan National Airport stations have in common: yellow line. 18. Begin at Shaw-Howard Univ station and follow the yellow line. 19. Count the number of stops until it reaches Ronald Reagan National Airport station. 20. Final answer: 8.

## Subtask Ambiguity Error Analysis

**Analysis**: The subtask asks how many subway stations there are between the stations, but the question actually requires calculating how many subway stations you need to pass to reach your destination, which is different.

**Agent Content**:
{ "subtask": "Calculate how many metro stations are **between the station nearest to Cleveland Elementary School and the station nearest to Fire Station 301 DCA ARFF**, without changing lines. Provide the final answer as a numerical integer value.",
"result": "Calculating Metro Stations Between Shaw-Howard University and National Airport Based on the information provided in the dependency results and my analysis of the Washington DC Metro system, I need to calculate how many metro stations are between the station nearest to Cleveland Elementary School and the station nearest to Fire Station 301 DCA ARFF, without changing lines. Identifying the Stations From the dependency results:- The station nearest to Cleveland Elementary School is **Shaw-Howard University** on the Green Line The station nearest to Fire Station 301 DCA ARFF is **Ronald Reagan Washington National Airport** (also called "National Airport") on the Blue and Yellow Lines Determining the Connecting LineAccording to the dependency results, the Yellow Line connects both stations. This means a person can travel between these two stations without changing lines by using the Yellow Line. Calculating the Number of StationsI analyzed the Yellow Line stations from north to south and found:- Shaw-Howard University is station 9 on the Yellow Line- National Airport is station 17 on the Yellow LineThe stations between them (excluding the starting station) are: 1. Mount Vernon Square 2. Gallery Place 3. Archives 4. L'Enfant Plaza 5. Pentagon 6. Pentagon City 7. Crystal City Therefore, there are **7** metro stations between Shaw-Howard University and National Airport on the Yellow Line without changing lines. Final Answer The number of metro stations between the station nearest to Cleveland Elementary School and the station nearest to Fire Station 301 DCA ARFF, without changing lines, is **7**.",
}

## Tool Selection Error: Error Case Level3 #8

**Question**: I was trying to remember how well the Cheater Beater performed in comparison to the Cheater when James tested it on his channel. I know that the Cheater still outperformed the Cheater Beater in terms of CFM. Could you please look that up for me, and report the CFM of both the Cheater and the Cheater Beater? I'm not sure if he made any changes to his testing, but this was back in season 4, so just report the value from that season. Please format your response like this: CFM number for Cheater, CFM number for Cheater beater.

**Answer**: 101.376, 84.348

**Annotation steps**: 1. Using a web browser, navigate to a search engine and conduct a search: "James Cheater Cheater Beater CFM Season 4" 2. Finding no relevant result, navigate to a search engine and conduct another search: "Cheater Beater Season 4" 3. Navigate to the first search result, https://www.youtube.com/watch?v=2vq3COPZbKo 4. Evaluate the YouTube page, noting that the video description identifies the video content comparing the performance of computer fans to a fan referred to as the "cheater" 5. Follow the link to the YouTube channel Major Hardware, https://www.youtube.com/@MajorHardware 6. Navigate to the About tab link, https://www.youtube.com/@MajorHardware/about 7. Evaluate the content, noting that the page identifies the operator of the channel as James 8. Navigate to a search engine and conduct a search, "James Major Hardware Cheater Beater" 9. Navigate to the first result, identical to the result from step 3 above, https://www.youtube.com/watch?v=2vq3COPZbKo 10. Search the page for CFM, finding no result 11. Load the video content and review it 12. Note an onscreen text element identifying a fan as "CALL SIGN: CHEATER BEATER" at timestamp 224 13. Note an onscreen table identifying the performance of various fans tested during season four, at timestamp 485 14. Evaluate the table content, identifying an entry for a fan named "Cheater" and a fan named "Cheater Beater" 15. Evaluate the table content, identifying that the data for both fans were recorded in season 4, S4E1 for Cheater, S4E6 for Cheater Beater 16. Record the data from the CFM column for the two fans, "Cheater: 101.376", and "Cheater Beater: 84.348" 17. Report the correct response to my user: "Cheater: 101.376 Cheater Beater: 84.348".

## Tool Selection Error Error Analysis

**Analysis**: The agent traverses season4's videos, but only stops at searching and does not open the video page for searching and processing.

**Agent Content**:
{ "role": "assistant", "tool_calls": [ {wq "type": "function",
"function": { "name": "**web_search**",
"arguments": "{"question": "Major Hardware Fan Showdown **Season 4 Episode 6 Cheater Beater**"}" } } ], },
{ "role": "assistant", "tool_calls": [ { "type": "function",
"function": { "name": "**web_search**",
"arguments": "{"question": "Major Hardware Fan Showdown Season 4 Episode 7 Cheater Beater"}" } } ], },
{ "role": "assistant", "tool_calls": [ { "type": "function",
"function": { "name": "web_search",
"arguments": "{"question": "Major Hardware Fan Showdown Season 4 Episode 8 Cheater Beater"}" } } ] }

---

## Web Toolkit Failure: Error Case Level1 #38

**Question**: On Cornell Law School website's legal information institute, under the fifth section of federal rules alphabetically, what word was deleted in the last amendment to the first rule in the article that has "witnesses" in the most titles as of 2021?

**Answer**: inference

**Annotation steps**: 1. Searched "Cornell Law School legal information institute" on Google. 2. Opened https://www.law.cornell.edu/. 3. Clicked Get The Law ¿ Federal Rules ¿ Federal Rules of Evidence (fourth section down). 4. Found the article that has "witnesses" in the most titles (VII). 5. Opened the first rule (701). 6. Scrolled to the last amendment as of 2021 (2011 amendment). 7. Found the word that was deleted (inference).

---

## Web Toolkit Failure Error Analysis

**Analysis**: The question asked to count the titles containing the exact word "witnesses". As a result, the browser also counted the titles containing the singular "witness" (such as "Who May Impeach a Witness").

**Agent Content**:
{ "role": "assistant",
"tool_calls": [ { "type": "function",
"function": { "name": "browse_url",
"arguments": ""start_url": "https://www.law.cornell.edu/rules/fre/article_vi", "task_prompt": "List all the rule titles in Article VI of the Federal Rules of Evidence to count how many contain the word 'witnesses'"" } } ] },
{ "role": "tool",
"content": "The task was to list all the rule titles in Article VI of the Federal Rules of Evidence and count how many contain the word "witnesses." Here's the final answer:The following rule titles contain the word "witnesses": 1. Rule 605. Judges Competency as a **Witness** 2. Rule 606. Jurors Competency as a **Witness** 3. Rule 607. Who May Impeach a **Witness** 4. Rule 608. A **Witness**s Character for Truthfulness or Untruthfulness 5. Rule 611. Mode and Order of Examining Witnesses and Presenting Evidence 6. Rule 612. Writing Used to Refresh a **Witness** 7. Rule 613. **Witness**s Prior Statement 8. Rule 614. Courts Calling or Examining a **Witness** 9. Rule 615. Excluding Witnesses. Total count: 9 rule titles contain the word "witnesses."", }

## Document Toolkit Failure: Error Case Level2 #44

**Question**: The work referenced in footnote 397 of Federico Lauria's 2014 dissertation is also the source for the titles of two paintings in the Smithsonian American Art Museum's collection, as of August 2023. What is the absolute difference between the chapter numbers of the chapters that the titles of these two paintings quote?

**Answer**: 8

**Annotation steps**: 1. Use search engine to search for "Federico Lauria's 2014 dissertation". 2. Open the result from philarchive.org and open the PDF file for the full paper. 3. Search for footnote 397 to find that the referenced work is Thomas Hobbes's "Leviathan". 4. Use search engine to search for "Smithsonian American Art Museum collection search". 5. Go to the museum's search webpage. 6. Enter "Hobbes Leviathan" into the search box and submit the search. 7. Open the two results, one by Jan Stussy ("A free man...") and one by Leon Karp ("Hereby it is manifest..."). 8. Verify from the full titles of these works that the titles are quotes from "Leviathan". 9. Use search engine to search for "Thomas Hobbes Leviathan full text". 10. Open any result that contains the full text, like the Project Gutenberg version. 11. Search the text for the titles of each painting, using different substrings from the titles as needed to account for variations in spelling and punctuation. 12. Find that the "A free man..." quote is from Chapter XXI (21) and that the "Hereby it is manifest..." quote is from Chapter XIII (13). 13. Calculate the absolute difference of the chapter numbers: 21 - 13 = 8.

## Document Toolkit Failure Error Analysis

**Analysis**: The PDF address of the paper has been determined, but the Page limit exceeded.

**Agent Content**:
{ "role": "assistant", "tool_calls": [ { "function": { "name": "extract_document_content", "arguments": "{ "document_path": "https://philarchive.org/go.pl?id=LAUQLO&proxyId=&u= https%3A%2F%2Fphilpapers.org%2Farchive%2FLAUQLO.pdf", "query": "footnote 397 Moby Dick chapter titles" }" } ] },
{ "role": "tool",
"content": "(True, **Error while processing document: Page limit exceeded**')" }

## Multimodal Toolkit Failure: Error Case Level3 #17

**Question**: Which of the fruits shown in the 2008 painting "Embroidery from Uzbekistan" were served as part of the October 1949 breakfast menu for the ocean liner that was later used as a floating prop for the film "The Last Voyage"? Give the items as a comma-separated list, ordering them in clockwise order based on their arrangement in the painting starting from the 12 o'clock position. Use the plural form of each fruit.

**Answer**: pears, bananas

**Annotation steps**: 1. Use search engine to search for "2008 painting Embroidery from Uzbekistan". 2. Open the top result, a link to the painting's page on the Dayton Art Institute website, and verify that the painting has the specified title and year. 3. Identify the fruits in the painting as watermelon, pear, lemon, and banana, which can be verified by either watching the video on the page or reading its linked transcript. 4. Use search engine to search for "ocean liner floating prop The Last Voyage". 5. Note from the results that this ocean liner was the SS Ŏ0cele de France. 6. Use search engine to search for "October 1949 breakfast menuSS Ŏ0cele de France". 7. Go to the result that shows the vintage SS Ŏ0cel de France breakfast menu for October 1949. 8. Search the menu for each of the four fruits from the painting, finding "Pear" and "Bananas" but no matches for "lemon" or "watermelon". 9. Check the positions of the fruits in the painting to find that the pears come before the bananas in clockwise order starting from the 12 o'clock position. 10. Format the final answer as specified using the correct ordering: pears, bananas.

## Multimodal Toolkit Failure Error Analysis

**Analysis**: ImageAnalysisToolkit recognizes that the banana is placed at 12 o'clock and the pear is placed at 9 o'clock, but their actual positions are 11 o'clock and 7 o'clock respectively.

**Agent Content**:
{ "subtask": "Create the final answer by listing the matching fruits in a comma-separated list, ordered clockwise from the 12 o'clock position as they appear in the painting, using the plural form of each fruit.",
"result": "I'll solve this task by creating a final answer that lists the matching fruits in a comma-separated list, ordered clockwise from the 12 o'clock position as they appear in the painting, using the plural form of each fruit. Based on the dependency results, I need to: 1. Identify which fruits from the painting "Embroidery from Uzbekistan" were also served on the SS le de France's October 1949 breakfast menu 2. List these matching fruits in clockwise order starting from the 12 o'clock position 3. Use the plural form of each fruit From the comparison analysis in the dependency results, I can see that the fruits that appear in both the painting and the breakfast menu are: - Bananas - Pears Now, I need to arrange these in clockwise order starting from the 12 o'clock position as they appear in the painting. According to the dependency results, the clockwise order of fruits in the painting starting from 12 o'clock is: **1. Bananas (12 o'clock) 2. Watermelon (3 o'clock) 3. Pears (6 o'clock) 4. Lemons (9 o'clock)** Since only bananas and pears match with the breakfast menu, and they are already in plural form, the final answer is: bananas, pears This is the comma-separated list of matching fruits in clockwise order from the 12 o'clock position as they appear in the painting, using the plural form of each fruit." }

## Response Format: Error Case Level2 #85

**Question**: At the two-minute mark in the YouTube video uploaded by the channel GameGrumps on May 14, 2017 as part of their playthrough of the game Mario Kart 8 Deluxe, the show hosts are competing on one of the games racetracks. What was the world record time for that track in the games 150cc mode as of June 7, 2023? Express your answer in minutes and seconds, rounding the seconds to the nearest hundredth, e.g. 1:01.001.

**Answer**: 1:41.614

**Annotation steps**: 1. Search the web for "gamegrumps mario kart 8 deluxe may 14 2017". 2. Click on the YouTube video result. 3. Navigate to two minutes into the video. 4. Scroll further back until I see the name of the racecourse, Yoshi Circuit. 5. Search the web for "mario kart 8 deluxe yoshi circuit world record 150cc". 6. Scroll down until I find a reliable world record listing site. 7. Navigate through the site until I find the record that meets the specified criteria. 8. Read the date the record was set to confirm that it applies to the question's specified date.

## Response Format Error Analysis

**Analysis**: This question requires rounding the seconds to the nearest hundred, such as 1:01.001, but the agent finally gives the result of **1:41.61**.

**Agent Content**:
{ "role": "assistant",
"content": "I need to analyze the results from all subtasks to determine the final answer to the original question.Looking at the results: 1. The first subtask identified the correct GameGrumps video from May 14, 2017: "Mario Kart 8 Deluxe: The Grand Prix - PART 7" (https://www.youtube.com/watch?v=nvaLkvUkW0w). 2. The second subtask determined that at the two-minute mark of this video, the hosts are racing on **Yoshi Circuit**. 3. The third subtask found that the world record time for Yoshi Circuit in 150cc mode as of June 7, 2023 was **1:23.964**. 4. The fourth subtask formatted this time as **1:23.96** (rounding to the nearest hundredth). 5.However, the verification task found a different time: **1:41.614** from an archived version of the Mario Kart 8 Deluxe World Records website from May 29, 2023. There's a discrepancy between the two reported world record times. The verification task provides more specific details, including: - The record holder: "Army" from France Date set: March 10, 2023

- Character/kart setup used: Birdo (Black) with Mr. Scooty kart, Azure Roller tires, and Paper Glider The verification task also specifically mentions "GCN Yoshi Circuit," which is the full name of the track (GCN indicating it originated in GameCube's Mario Kart: Double Dash).Given that the verification task provides more detailed information and cites a specific archive date (May 29, 2023) that is closer to our target date (June 7, 2023), I will consider this the more reliable result.Therefore, **the final answer to the original question is: 1:41.61**(1 minute, 41.61 seconds)This is the world record time for Yoshi Circuit in Mario Kart 8 Deluxe's 150cc mode as of June 7, 2023, expressed in minutes and seconds with the seconds rounded to the nearest hundredth." }

## Reasoning Error: Error Case Level3 #25

**Question**: In NASA's Astronomy Picture of the Day on 2006 January 21, two astronauts are visible, with one appearing much smaller than the other. As of August 2023, out of the astronauts in the NASA Astronaut Group that the smaller astronaut was a member of, which one spent the least time in space, and how many minutes did he spend in space, rounded to the nearest minute? Exclude any astronauts who did not spend any time in space. Give the last name of the astronaut, separated from the number of minutes by a semicolon.

**Answer**: White; 5876

**Annotation steps**: 1.Use search engine to search for "NASA's Astronomy Picture of the Day 2006 January 21". 2. Open the link to the image. 3. Read the explanation to find that the image is of astronaut Charles "Pete" Conrad reflected in the helmet of astronaut Alan Bean. 4. Observe that the smaller astronaut in the image is the one reflected in the other's helmet, so the smaller astronaut must be Charles "Pete" Conrad. 5. Go to the Wikipedia page for Charles "Pete" Conrad. 6. Search for "Astronaut Group" to find that Conrad was a member of NASA Astronaut Group 2. 7. Open the Wikipedia pages for each member of NASA Astronaut Group 2. 8. For those who are not deceased, go to View history and select the latest version of their Wikipedia page as of August 2023. 9. Compare the times listed in the infobox of each astronaut's Wikipedia page under "Time in space", observing that Ed White has the least time in space with 4d 01h 56m, but also that Elliott See does not have a listed "Time in space". 10. Read through Elliott See's Wikipedia article to find that he died in an accident before his first space flight, so he should be excluded, making Ed White's 4d 01h 56m the least amount of time in space. 11. Convert 4d 01h 56m to minutes: 4d * 24h/d * 60m/h + 1h * 60m/h + 56m = 5,876m. 12. Format the final answer as specified: White; 5,876.

## Reasoning Error Error Analysis

**Analysis**: The accumulated time in space for NASA's "New Nine" had been correctly captured, but the people who had not been to space were incorrectly taken into account.

**Agent Content**:
{ "subtask": "Research the space time duration for each astronaut in this group. Use web search to find reliable sources about each astronaut's missions and total time spent in space. Create a comprehensive list with their names and space time durations.",
"result": "Space Time Durations for NASA Astronaut Group 2 Members"
1. **Neil Armstrong** Total time in space: 8 days, 14 hours, 12 minutes Missions: Gemini 8 (1966), Apollo 11 (1969)
2. **Frank Borman** Total time in space: 19 days, 21 hours, 35 minutes Missions: Gemini 7 (1965), Apollo 8 (1968)
3. **Charles Pete Conrad** Total time in space: 49 days, 3 hours, 38 minutes Missions: Gemini 5 (1965), Gemini 11 (1966), Apollo 12 (1969), Skylab 2 (1973)
4. **Jim Lovell** Total time in space: 29 days, 19 hours, 5 minutes (715 hours, 5 minutes) Missions: Gemini 7 (1965), Gemini 12 (1966), Apollo 8 (1968), Apollo 13 (1970)
5. **James McDivitt** Total time in space: 14 days, 2 hours, 56 minutes Missions: Gemini 4 (1965), Apollo 9 (1969)
6. **Elliot See** Total time in space: 0 minutes Note: Selected but never flew; died before Gemini 9 mission.

7. **Thomas P. Stafford**  Total time in space: 21 days, 3 hours, 43 minutes (507 hours, 43 minutes)  Missions: Gemini 6A (1965), Gemini 9A (1966), Apollo 10 (1969), Apollo-Soyuz Test Project (1975)
8. **Ed White**  Total time in space: 4 days, 1 hour, 56 minutes  Missions: Gemini 4 (1965); first American spacewalk; died in Apollo 1 fire.
9. **John Young**  Total time in space: 34 days, 19 hours, 39 minutes  Missions: Gemini 3 (1965), Gemini 10 (1966), Apollo 10 (1969), Apollo 16 (1972), STS-1 (1981), STS-9 (1983) }

## Language/Question Ambiguity: Error Case Level2 #73

**Question**: I'm curious about how much information is available for popular video games before their release. Find the Wikipedia page for the 2019 game that won the British Academy Games Awards. How many revisions did that page have before the month listed as the game's release date on that Wikipedia page (as of the most recent entry from 2022)?

**Answer**: 60

**Annotation steps**: 1. Search the web for British Academy Video Games Award for Best Game 2019. 2. Find the answer, Outer Wilds. 3. Find the Wikipedia page for Outer Wilds. 4. Go to the last revision from 2022. 5. Note the release date, May 29, 2019. 6. View the page history. 7. Count how many edits were made to the page before May 2019. 8. Arrive at the answer, 60.

## Language/Question Ambiguity Error Analysis

**Analysis**: There is an ambiguity in interpreting the phrase 2019 game that won the British Academy Games Awards. One interpretation refers to the game that won the BAFTA Best Game award at the 2019 ceremony, regardless of its actual release year. The other interpretation requires the game to have been both released in 2019 and won the BAFTA Game Award. This difference leads to inconsistent results, as the agent prefers the former (based solely on the award year), while the ground truth requires both conditions (release year and award) to be met.

**Agent Content**:
{"subtask": "Search for the **2019 game that won the British Academy Games Awards (BAFTA)** and identify the specific game.",
"result": "2019 BAFTA Games Awards Winner God of War Based on the information from the 15th British Academy Games Awards held on April 4, 2019, **God of War** (2018) was the winner of the Best Game award. Game Details - **Developer**: Santa Monica Studio - **Publisher**: Sony Interactive Entertainment Europe - **Platform**: PlayStation 4 - **Release Date**: April 2018 Awards Won (5 total) 1. Best Game 2. Audio Achievement 3. Music 4. Narrative 5. Performer (Jeremy Davies for his role as T̈he Stranger) Additional Information - God of War received 10 nominations in total, making it the most nominated game at the ceremony - The 15th BAFTA Games Awards honored games released in 2018, though the ceremony was held in 2019"}

## Hallucination: Error Case Level2 #79

**Question**: If this whole pint is made up of ice cream, how many percent above or below the US federal standards for butterfat content is it when using the standards as reported by Wikipedia in 2020? Answer as + or - a number rounded to one decimal place.

**Answer**: +4.6

**Annotation steps**: 1. Open the image. 2. Search "butterfat wikipedia" on Google search. 3. Open the Butterfat Wikipedia page. 4. Click "View history" on the page. 5. Scroll down to the end of 2020 and click the last 2020 version of the page. 6. Check the ice cream requirement for fat content (10%). 7. Click "View history" on the page. 8. Scroll down to the beginning of 2020 and click the last 2019 version of the page. 9. Check the ice cream requirement for fat content to ensure it's the same (10%). 10. Calculate the fat percentage of the pint of ice cream from the image of the nutrition panel (21g fat per serving / 144g ice cream per serving = 14.6%). 11. Calculate the difference from the standard (14.6% - 10% = 4.6%).

## Hallucination Error Analysis

**Analysis**: The agent subjectively assumes that 85% of the total fat is milk fat to estimate the actual milk fat content, but the question does not provide additional information.

**Agent Content**:
{ "subtask": "Determine the actual butterfat content of the ice cream shown in the image or described in the task.",
"result": "Based on my analysis, I have determined the actual butterfat content of the ice cream shown in the image. Methodology 1. First, I examined the nutrition facts on the ice cream container, which shows: - Serving size: 2/3 cup (144g) - Total fat per serving: 21g 2. I calculated the total fat percentage in the ice cream: - (21g fat 144g serving) 100 = 14.6% total fat 3. I analyzed the ingredients list to determine what portion of this fat is butterfat: - Cream is the first ingredient (highest proportion by weight) - Other dairy fat sources include butter and butteroil - Non-dairy fat sources include coconut oil and soybean oil 4. Based on the ingredients list with cream as the first ingredient and the presence of butter and butteroil, **I estimated that approximately 85% of the total fat is butterfat**, which is a standard industry estimate for premium ice cream with these ingredients. Result The estimated butterfat content of this ice cream is 12.4%. This was calculated as: - 14.6% (total fat) 0.85 (estimated butterfat proportion) = 12.4% To account for uncertainty, I also calculated a possible range: - Lower estimate: 11.7% (assuming 80% of fat is butterfat) - Higher estimate: 13.1% (assuming 90% of fat is butterfat) The final butterfat content of 12.4% is my best estimate based on the available information from the nutrition facts and ingredients list." }

## Context Exceed: Error Case Level3 #5

**Question**: What animals that were mentioned in both Ilias Lagkouvardos's and Olga Tapia's papers on the alvei species of the genus named for Copenhagen outside the bibliographies were also present in the 2021 article cited on the alvei species' Wikipedia page about a multicenter, randomized, double-blind study?

**Answer**: mice

**Annotation steps**: 1. Searched "alvei copenhagen" on Google. 2. Opened https://en.wikipedia.org/wiki/ Hafnia_(bacterium). 3. Searched "Ilias Lagkouvardos hafnia alvei" on Google. 4. Opened https://www.mdpi.com/2076-2607/11/1/123?type=check_update&version=2. 5. Opened a new tab. 6. Searched "Olga Tapia hafnia alvei" on Google. 7. Opened https://pubmed.ncbi.nlm.nih.gov/36080356/. 8. Found all animals mentioned in the first paper. 9. Searched each animal from the first paper in the second paper. 10. Noted the animals mentioned in both outside the bibliographies. 11. Went back to the Wikipedia article. 12. Opened the link in the references to "The Probiotic Strain H. alvei HA4597® Improves Weight Loss in Overweight Subjects under Moderate Hypocaloric Diet: A Proof-of-Concept, Multicenter Randomized, Double-Blind Placebo-Controlled Study". 13. Opened the PDF. 14. Found the animals shared by all three papers.

## Context Exceed Error Analysis

**Analysis**: The content of Olga Tapiaresearch has been extracted correctly, but the mice mentioned in the text have not been extracted from the content successfully.

**Agent Content**:
{ "role": "assistant",
"tool_calls": [ {
"type": "function",
"function": {
"name": "extract_document_content",
"arguments": "{"document_path": "https://www.mdpi.com/1420-3049/27/17/5585", "query": "Olga Tapia research on Hafnia alvei with mentions of animals"}"
} } ] }

## Limited Coding Capability: Error Case Level1 #36

**Question**: Bob was invited to participate in a game show, and he advanced to the final round. The final round offered Bob the chance to win a large sum by playing a game against the host. The host has 30 shiny prop coins, each of which is worth $1,000 if Bob manages to win them by playing the game. The host hides the coins in three different prize boxes and then shuffles their order. The only rule restricting the host's coin placement is that one box must contain at least 2 coins, and one box must contain 6 more coins than another box. In order to play, Bob must submit three guesses, one guess for the number of coins in each box. The box is then opened and the number of coins is revealed. If Bob's guess is a number greater than the number of coins in the box, Bob earns no coins. If Bob guesses a number equal to or less than the number of coins in the box, Bob wins a number of coins equal to his guess.If Bob plays uses the optimal strategy, what's the minimum amount of money he can win from the game?

**Answer**: 16000

**Annotation steps**: Step 1: Evaluate the problem statement provided by my user, storing the relevant information: 30 coins with a value of $1,000 distributed between 3 boxes. Each box must contain at least 2 coins. One box must contain 6 more coins than another. Step 2: Evaluate the base distribution: 2-8-20, noting that two boxes must contain at least 8 coins. Step 3: Evaluate the most even allowable distribution: 8,8,14, noting that two boxes must contain at least 8 coins. Step 4: Evaluate a case where Bob guesses 8 for each box in the outlier distributions. Step 5: For the worst case 2-8-20 distribution, Bob wins 0+8+8 = 16 coins. Step 6: For the 8-8-14 distribution, Bob wins 8+8+8 = 24 coins. Step 7: Convert the worst-case coin count to a prize value, 16*$1,000 = $16,000. Step 8: Report the correct answer to my user: "$16,000"

## Limited Coding Capability Error Analysis

**Analysis**: The agent ignored the hard constraint that each box must contain at least 2 coins when writing the code, and mistakenly considered the distribution of 0 and 1 coins (such as (0,6,24), (1,7,22), etc.) as legal.

**Agent Content**:
{"subtask": "Implement a solution using code to systematically explore all valid coin distributions, calculate Bob's optimal guesses for each distribution, and determine the minimum guaranteed winnings.",
"result": "Optimal Strategy for the Game Show Problem I've implemented a solution to systematically explore all valid coin distributions, calculate Bob's optimal guesses for each distribution, and determine the minimum guaranteed winnings. Let me explain my approach and findings. Solution Approach 1. **Generate Valid Distributions**: I created a function to generate all valid distributions of 30 coins among 3 boxes that satisfy the constraints: - Total of 30 coins - **One box must contain at least 2 coins** - One box must contain 6 more coins than another box 2. **Calculate Winnings**: For each distribution and guessing strategy, I calculated the maximum possible winnings by trying all permutations of the guesses (since Bob can choose which guess to apply to which box). 3. **Find Optimal Strategy**: I implemented a systematic search to find the strategy that maximizes the minimum guaranteed winnings across all possible distributions. Key Findings 1. **Valid Distributions**: The code identified 12 valid distributions of 30 coins that satisfy all constraints. 2. **Optimal Strategy**: **The optimal strategy for Bob is to guess **[0, 6, 12]** for the three boxes.** 3. **Minimum Guaranteed Winnings**: With this strategy, Bob is guaranteed to win at least **$18,000**, regardless of how the host distributes and shuffles the coins. 4. **Worst Case Scenario**: The worst-case distribution for Bob is **(0, 6, 24)**, where one box has 0 coins, one has 6 coins, and one has 24 coins. 5. **Verification**: I verified that with the optimal strategy against the worst-case distribution, Bob indeed wins exactly $18,000. Comparison with Other Strategies I also tested several other strategies mentioned in the dependency results: - Strategy (6, 6, 6): Minimum guaranteed $12,000 - Strategy (2, 6, 10): Minimum guaranteed $16,000 - Strategy (1, 5, 12): Minimum guaranteed $17,000 - Strategy (7, 7, 7): Minimum guaranteed $7,000 - Strategy (8, 8, 8): Minimum guaranteed $8,000 - Strategy (6, 6, 7): Minimum guaranteed $13,000 - Strategy (6, 7, 7): Minimum guaranteed $13,000 This confirms that [0, 6, 12] is indeed the optimal strategy, guaranteeing $18,000 in winnings. How the

Optimal Strategy Works With the [0, 6, 12] strategy: 1. Bob guesses 0 for the box with the fewest coins 2. Bob guesses 6 for the box with the middle number of coins 3. Bob guesses 12 for the box with the most coins This strategy works because: - It ensures Bob always wins from at least two boxes - Even in the worst-case distribution (0, 6, 24), Bob can match his guesses optimally to win $18,000 - No other strategy can guarantee more than $18,000 across all possible distributions Therefore, $18,000 is the minimum amount Bob can win using the optimal strategy." }

# F   Case Studies

The real environment is often more complex than the pre-built simulation environment. The GAIA benchmark is built in real-life environments, which brings many challenges. Below we will introduce the challenges that agents may face in solving problems in real environments.

## F.1   Differences in Information Sources

In real-world scenarios, multiple information sources may provide different answers to the same query. This discrepancy creates a challenge where tasks might have several valid responses, each considered "correct" depending on the reference source consulted.

To illustrate this phenomenon, consider the following task:

---

**Task from GAIA Benchmark**

**Question**: Of the authors (First M. Last) that worked on the paper "Pie Menus or Linear Menus, Which Is Better?" in 2015, what was the title of the first paper authored by the one that had authored prior papers?

**Answer**: Mapping Human Oriented Information to Software Agents for Online Systems Usage.

**Annotation steps**: 1. Searched "Pie Menus or Linear Menus, Which Is Better?" on Google. 2. Opened "Pie Menus or Linear Menus, Which Is Better?" on https://oda.oslomet.no/oda-xmlui/handle/10642/3162. 3. Clicked each author's name. 4. Noted the name that had no other papers listed. 5. Searched "Murano, Pietro" on Google. 6. Opened http://www.pietromurano.org/. 7. Clicked "Publications". 8. Found the earliest paper he contributed to.

---

as shown in Figure 5, consulting the author's personal homepage yields "Mapping Human-Oriented Information to Software Agents for Online Systems Usage" as the answer. However when referencing Google Scholar, the first publication listed is "A new software agent 'learning' algorithm" as shown in Figure 6. This inconsistency demonstrates how different authoritative sources can lead to divergent yet equally defensible answers.

## F.2   Outdated information

**Outdated information** is another common issue, especially when retrieving data via web search and information retrieval tools. The agent might encounter outdated or incorrect content that misguides its reasoning process, ultimately impacting the quality and effectiveness of task execution. For example, the original task may require visiting someone's YouTube channel or searching for some content in the channel. However, by now, the content may have been modified by the channel user, which will directly lead to error in the ground truth.

Here is an example from GAIA benchmark:

Murano, Pietro (2003) Anthropomorphic Vs Non-Anthropomorphic Software Interface Feedback for Online Factual Delivery - PDF, 7th International Conference on Information Visualisation (IV 2003) An International Conference on Computer Visualisation and Graphics Applications, London, England, 16-18 July 2003, (c) - IEEE.

Murano, Pietro (2002) Anthropomorphic Vs Non-Anthropomorphic Software Interface Feedback for Online Systems Usage - PDF, 7th European Research Consortium for Informatics and Mathematics (ERCIM) Workshop - 'User Interfaces for All' - Special Theme: 'Universal Access'.Paris(Chantilly),France 24,25 October 2002. Published in Lecture Notes in Computer Science (C) - Springer.

Murano, Pietro (2002) Effectiveness of Mapping Human-Oriented Information to Feedback From a Software Interface - PDF, Proceedings of the 24th International Conference on Information Technology Interfaces, Cavtat, Croatia, 24-27 June 2002.

Murano, Pietro (2001) A New Software Agent 'Learning' Algorithm - PDF, People in Control An International Conference on Human Interfaces in Control Rooms, Cockpits and Command Centres, UMIST, UK, IEE.

Murano, Pietro (2001) Mapping Human-Oriented Information to Software Agents for Online Systems Usage - PDF People in Control An International Conference on Human Interfaces in Control Rooms, Cockpits and Command Centres, UMIST, UK, IEE.

## INTERNATIONAL PRESENTATIONS

Ritchings, Tim. Papoulakis, George. Garvey, David. Murano, Pietro. Sheta, Walaa. (2003) Augmenting Environments By Automated Sign Location and Text Detection - PDF HCI International 2003, 10th International Conference on Human Computer Interaction - Symposium on Human Interface (Japan) 2003 - 5th International Conference on Engineering Psychology and Cognitive Ergonomics - 2nd International Conference on Universal Access in Human Computer Interaction. 22-27 June 2003, Crete, Greece.

Effectiveness of Mapping Human-Oriented Information to Software Agents for Online Teaching Environments Womens Engineering Society (WES) 73rd Annual Conference - Human Interface With Machines. UCLAN,UK, September 2000.

Figure 5: Pietro Murano's personal homepage showing publication history (accessed on 2025-05)

| | | |
|---|---|---|
| Part IV-Novel Interaction Paradigms--New Modalities and Dialogue Styles-Anthropomorphic vs. Non-anthropomorphic Software Interface Feedback for Online Systems Usage
P Murano
Lecture Notes in Computer Science 2615, 339-349 | | 2003 |
| Anthropomorphic vs. non-anthropomorphic software interface feedback for online systems usage
P Murano
ERCIM Workshop on User Interfaces for All, 339-349 | 10 | 2002 |
| Anthropomorphic Vs Non-Anthropomorphic Software Interface Feedback for Online Systems Usage, 7th European Research Consortium for Informatics and Mathematics (ERCIM) Workshop …
P Murano
Paris (Chantilly), France 24, 25 | 6 | 2002 |
| Effectiveness of mapping human-oriented information to feedback from a software interface
P Murano
ITI 2002. Proceedings of the 24th International Conference on Information … | 16 | 2002 |
| Mapping human-oriented information to software agents for online systems usage
P Murano
2001 People in Control. The Second International Conference on Human … | 10 | 2001 |
| A new software agent'learning'algorithm
P Murano
2001 People in Control. The Second International Conference on Human … | 16 | 2001 |
| Effectiveness of VRML Building Models for Direction Finding
P Murano, G Manchester, D Mackey | | |

Figure 6: Pietro Murano's Google Scholar profile displaying a different publication order (accessed on 2025-05)

> **Task from GAIA Benchmark**
>
> **Question**: Eva Draconis has a personal website which can be accessed on her YouTube page. What is the meaning of the only symbol seen in the top banner that has a curved line that isn't a circle or a portion of a circle? Answer without punctuation.
>
> **Answer**: War is not here this is a land of peace.
>
> **Annotation steps**: 1. By googling Eva Draconis youtube, you can find her channel. 2. In her about section, she has written her website URL, orionmindproject.com. 3. Entering this website, you can see a series of symbols at the top, and the text "see what the symbols mean here" below it. 4. Reading through the entries, you can see a short description of some of the symbols. 5. The only symbol with a curved line that isn't a circle or a portion of a circle is the last one. 6. Note that the symbol supposedly means "War is not here, this is a land of peace."

According to the annotation, the agent should first go to Eva Draconis youtube channel, and check her personal website URL from about section. However, there's nothing in the channel now, and the personal webpage cannot be opened either.

### F.3 Language Ambiguity

**Language ambiguity** occurs when the agent encounters unclear or imprecise language in the user query, leading to incorrect reasoning. For example, a question might contain terms with multiple meanings or lack the necessary context to disambiguate the intended meaning.

For example, consider the following task:

> **Task from GAIA Benchmark**
>
> **Question**: I'm curious about how much information is available for popular video games before their release. Find the Wikipedia page for the 2019 game that won the British Academy Games Awards. How many revisions did that page have before the month listed as the game's release date on that Wikipedia page (as of the most recent entry from 2022)?
>
> **Answer**: 60
>
> **Annotation steps**: 1. Search the web for British Academy Video Games Award for Best Game 2019 2. Find the answer, Outer Wilds 3. Find the Wikipedia page for Outer Wilds 4. Go to the last revision from 2022. 5. Note the release date, May 29, 2019 6. View the page history 7. Count how many edits were made to the page before May 2019 8. Arrive at the answer, 60

There are two possible interpretations of the question: (1) "2019 award winner": refers to the game that won the "Best Game" award at the 2019 BAFTA Games Awards (British Academy Games Awards), regardless of when it was actually released. (2) "2019 released and won the game": refers to the game that was released in 2019 and has won the BAFTA Game Award.

### F.4 Network Instability & Permission Deny

**Network instability** is a critical factor impacting task evaluation. Since web search tools rely on internet connection, network instability can result in the agent being unable to access required resources or cause task execution to be interrupted.

**Permission deny** is also a critical factor, which refers to scenarios where the agent attempts to access resources that require (1) authentication or (2) human verification (such as downloading a paper from a journal that requires login access or human verification). In some cases, IP region restriction could also prevent agent from futher processing the task. This could cause the agent to fail in completing the task, even if it has found the correct URL containing the final answer.

# G   Broader Impacts

## G.1   Data Pollution

**Data pollution** refers to a situation where the agent might directly retrieve answers from a source (such as GAIA's Huggingface) instead of performing its own reasoning. This issue may lead to the agent "cheating" by relying on pre-existing answers from the training data, rather than conducting the reasoning required for the task. A simple way to solve this problem is restrict the agent's access to certain sources (e.g., by filtering out urls containing "huggingface" to prevent the agent from retrieving answers directly from specific datasets).

## G.2   Potential Dangerous Behaviors

While using tools, the agent might display **dangerous behaviors**, such as accessing harmful content, pushing inaccurate information, generating inappropriate interactions, etc. If the agent can execute code/use the terminal locally, some operations of the agent may destroy some of the original local environment. These behaviors can pose a threat to user safety and system stability.

# H   Statistical Significance Analysis

To validate the robustness of our empirical findings, we conducted Wilcoxon signed-rank tests to compare system-level performance across key baselines and planner configurations.

First, we observe that WORKFORCE significantly outperforms both the Single Agent and Role Playing baselines, with $p$-values of $< 0.0001$ and $0.0203$ respectively. These results confirm that the observed improvements are statistically significant rather than due to random variability.

Furthermore, we evaluate the impact of our proposed training method, *Optimized Workforce Learning (OWL)*, by comparing OWL-trained Qwen2.5-32B-Instruct against the base Qwen2.5-32B-Instruct model. The Wilcoxon test yields a $p$-value of $0.0018$, indicating a significant improvement in task planning effectiveness after OWL-based reinforcement learning.

# I   More Details on Analysis

## I.1   More Details on Robustness

| Model | Number of Capabilities | | | Mean ↑ | Std ↓ | IQR ↓ |
|---|---|---|---|---|---|---|
| | **1** | **2** | **≥3** | | | |
| Single Agent | 38.36 | 35.14 | 44.44 | 39.31 | 3.86 | 4.65 |
| Role Playing | 62.34 | 51.14 | 34.62 | 49.36 | 11.39 | 13.86 |
| Workforce (Ours) | 63.01 | 59.46 | 55.56 | 59.34 | 3.05 | 3.73 |

Table 9: Performance across different capability requirements, as well as overall robustness indicators including standard deviation and interquartile range (IQR)

The complete table of performance across different capability requirements is presented in Table 9.

## I.2   Planner vs. Worker Training

Table 10 presents a detailed comparison of different training configurations for our multi-agent system. The results clearly demonstrate that training only the planner (45.45% average accuracy) significantly outperforms training only the workers (31.51%), and even performs comparably to training both components together (46.68%). Interestingly, training only workers actually degrades performance below the baseline (36.36%), suggesting that worker specialization without proper task decomposition can be counterproductive. These findings strongly support our design decision to prioritize planner optimization, as effective task planning and decomposition are more crucial than enhancing individual worker capabilities. The minimal performance gain from training both

| Variant | Trainable Components | | Level 1 | Level 2 | Level 3 | Average |
|---|---|---|---|---|---|---|
| | **Planner** | **Worker** | | | | |
| Qwen-32B-Instruct | × | × | 49.05 | 33.72 | 19.23 | 36.36 |
| *w.* Worker | × | ✓ | 43.39 | 30.23 | 11.54 | 31.51 |
| *w.* Both | ✓ | ✓ | 60.38 | 47.67 | 15.38 | 46.68 |
| *w.* Planner (Ours) | ✓ | × | 60.38 | 45.34 | 15.38 | 45.45 |

Table 10: Performance comparison between different training configurations of planner and workers. Training the planner alone achieves nearly the same performance as training both components, while training only workers degrades performance.

components (only +1.23%) does not justify the substantial increase in computational costs, making planner-focused training the most efficient approach for improving overall system performance.

