# OpenReview forum: "OWL: Optimized Workforce Learning for General Multi-Agent Assistance in Real-World Task Automation"
_NeurIPS.cc/2025/Conference — NeurIPS 2025 poster_

### Official Review · Reviewer_vyod · 2025-06-30

**Clarity:** 3
**Significance:** 3
**Originality:** 3
**Rating:** 5
**Confidence:** 3

**Summary:**

This paper proposes a hierarchical method for multi-agent task execution using LLMs. The novelty of the work is that it uses a domain-agnostic planner and coordinator, which decomposes tasks and relays them to different agents, and domain-specific agent experts, which execute each part of the task. This enables to train only the experts on domain-specific data and maintain the same planner across domains and tasks. To train this system, the paper proposes OWL, which trains the domain agnostic planner via SFT on demonstratiosn from GPT-4o and RL on task success. Without any training, the approach obtains SOTA on the GAIA benchmark compared to other Open-Source frameworks, and is the first to outperform the closed-course Deep Research model. The training process also shows significant improvements, leading to 13% relative improvement with SFT and 46% relative improvement with DPO.

**Questions:**

In order of importance:
- Clarify differences between results in Table 2 and Table 3. Particularly difference between  Table 3,row 3 different from those on Table 2, last row.
- It would be great to test how the finetuned planner generalizes to unseen workers, and in particular whether there are still improvements wiht respect to the zero-shot version.
- Why only training the high-level planner? Would there be gains from training the coordinator as well?
- Clarify differences (besides the finetuning approach) with respect to works like Magnetic-One.
- Results: if SFT is finetuned on GPT4o-mini, why is there such a significant gap between the first and second to last row in in Table 3.

**Ethical Concerns:**

["NO or VERY MINOR ethics concerns only"]

**Final Justification:**

The rebuttal clarified my remaining doubts, I am keeping my score as I believe this paper should be accepted

**Limitations:**

Yes

**Quality:**

3

**Strengths And Weaknesses:**

Strengths:
- Relevance: The paper addresses an important problem, which has gained important in the research community: performing complex reasoning tasks by orchestrating multiple agents.
- Novelty: to my knowledge, this is the first work that learns a high level planner with task reward data, rather than training the actor agents. This seems like a powerful approach since there may be agents in the that cannot be finetuned (either because the provider does not allow it or because they are non-learned agents). This framework focuses on the high level planner which is agnostic to the agents and tasks. the finetuning approach obtains significant gains over the zero-shot approach.
- Experimental details: Provide a detailed analysis on the pretraining datasets used to improve the high-level planner, motivating the selection of different datasets and describing an approach to filter low-quality data from the experts, with significant gains on SFT. It would be great to look at the contribution of each dataset in GAIA performance.
- The proposed appraoch can be combined with different base models and obtains SOTA results on GAIA when compared to similar models. In the zero-shot setting, WorkForce with GPT-4o and Claude-3.7-Sonnet, outperforms all the other open source frameworks with similar base models by a wide margin, and are close to other closed-source frameworks. In the fine-tuning setting, the framework outperforms closed-source based models such as GPT-4o-mini and larger models.

Weaknesses:
- Clarity: The structure of the paper makes it somewhat hard to follow, there is a high overlap between preliminary and related work, and they could be moved as a single section. Similarly the results in section 3.3 and 4.4 should be comparable since the only difference seems to be on whether the Workforce approach is further pretrained or not. Making these in separate sections creates confusion on whether the evaluations (dataset metrics and baselines) are comparable or not.
- Results: if SFT is finetuned on GPT4o-mini, why is there such a significant gap between the first and second to last row in in Table 3.
- One of the main claims of the work is that the planner is agnostic to tasks and agents, and therefore can generalize to different types of agents. It would be great to test how overfit is the high-level planner to the specialized workers. What happens if you change the capacity of the workers or their specialization? Do you still get gains on the finetuned planner?
- Related work: It would be great to clarify the novelty with respect to related work, particularly what is the difference between the orchestration in your work vs Magnetic-One? You have a special agent that is doing task decomposition and one that is doing task planning, which seems similar to the orchestrator in Magnetic-One.
- Method and Clarity: The way the results are presented, I am confused about how table 3 compares to table 2. Are all results on Table 3 showing the workforce framework with different base planners? Why are the numbers for Claude-3.7-Sonnet in Table 3,row 3 different from those on Table 2, last row? Shouldn't it be the same method? It would be useful to see how the same exact planner model performs both when using the Workforce framework vs not. Ideally taking one of the frameworks from Table 2 and replacing them with the Qwen2.5-32B-Instruct base, this way it would be possible to see how much gain there is from the workforce framework vs the post-training.
- While the approach is promising, it looks like one big limitation is in the necessity to define worker agents per task. The result of this is that, for new tasks, it is still necesary to write agents with domain-specific knowledge. It would be great if authors could comment on the details of how these workers were defined in the experimental section and how scalable this is to new tasks.

---

> ### Author Rebuttal · Authors · 2025-07-31
>
> We appreciate your consideration of our work as an innovative solution to an important problem in the multi-agent field concerning complex reasoning tasks. Due to overlapping areas between some of the weaknesses and questions, we have merged the overlapping parts in our response to stay within the character limit.
>
> **W1. Clarity: The structure of the paper makes it somewhat hard to follow, there is a high overlap between preliminary and related work, and they could be moved as a single section. Similarly the results in section 3.3 and 4.4 should be comparable since the only difference seems to be on whether the Workforce approach is further pretrained or not. Making these in separate sections creates confusion on whether the evaluations (dataset metrics and baselines) are comparable or not.**
>
> Thank you for the suggestion. We will revise the structure to clarify the comparability between sections and reduce redundancy between the preliminary and related work.
>
> The reason we separated Sections 3.3 and 4.4 is that we aimed to highlight the individual contributions of the Workforce agent architecture and the OWL training method. While it's true that the two experimental tables could technically be merged—though at the cost of readability—the focus of each experiment differs in terms of conclusions, experimental settings, and implementation details. Keeping them separate allows us to clearly articulate the specific gains from architecture versus training improvements.
>
>
> **Q1.& W5. Clarify differences between results in Table 2 and Table 3. Particularly difference between Table 3,row 3 different from those on Table 2, last row.**
>
> Thank you for your question and for pointing out the need for clarification. We believe your concern refers to the experimental settings related to Tables 1 and 3 (since tables 2 and 3 are quite different). As noted in the caption of Table 3, this is a controlled experiment where the workers consistently use GPT-4o as their base model, while we vary the planner’s base model to examine its impact. In contrast, the last row of Table 2 corresponds to a different setting where both the planner and the workers use Claude-3.7-SONNET as their base model, as specified in Table 1. We will make this distinction clearer in the final version of the paper.
>
> **Q2.& W3. (1) It would be great if authors could comment on the details of how these workers were defined in the experimental section**
>
> As for the worker definition, our worker design adheres to two key principles: capability specialization and clear role definition. First, we minimize functional overlap between workers to ensure efficient task allocation and avoid redundancy. Second, we establish precise capability boundaries for each worker type to enable optimal decision-making by the Coordinator and Planner.
> In the current instantiation of Workforce, we primarily target general-purpose tasks, which demand agents equipped with at least the following core capabilities:
> (1) web browsing for information retrieval,
> (2) multimodal file processing for diverse input handling, and
> (3) advanced reasoning for complex problem-solving.
>
> **Q2.& W3. (2) how scalable this is to new tasks: how the finetuned planner generalizes to unseen workers, and in particular whether there are still improvements wiht respect to the zero-shot version.**
>
> We conducted a supplementary experiment to verify the generalization of the trained planner. Specifically, we added an additional agent for medical image analysis in medical scenarios. We used the dev dataset from the MedMCQA[1] dataset and tested the planner before and after training (based on the Qwen2.5-32b-Instruct model). The experimental results showed an accuracy of **71% before training** and **75% after training**, demonstrating that the trained planner maintains generalization capabilities in out-of-domain scenarios.
>
>
> **Q3. Why only training the high-level planner? Would there be gains from training the coordinator as well?**
>
> Thanks for your valuable concern about the coordinator. In our initial experience, the benefits of using models with different capability levels for the coordinator are not very different. Therefore, we didn't choose to train the coordinator. To verify this, we conducted a simple experiment to evaluate the impact of different base models on the coordinator to assign workers. Specifically, we used **GPT-4o** to generate subtasks for each of the GAIA tasks, with 783 subtasks in total. We then fixed the worker set and **manually labeled** the workers that should be used for each subtask, marking them as **ground truth**. We used **GPT-4o and GPT-4o-mini** as coordinators, respectively, and analyzed the **accuracy of task assignment** using different base models. The experimental results showed that **GPT-4o** achieved an accuracy of **88.8%** and **4o-mini** achieved an accuracy of **90.1%**, which means that when the boundaries between different workers are clear enough, an LLM with general capabilities can already assign the correct worker to the current task well.
>
>
> **Q4.& W4. Clarify differences (besides the finetuning approach) with respect to works like Magnetic-One.**
>
> We sincerely appreciate this opportunity to clarify the difference between our work and Magentic-One. Beyond fine-tuning, our system differs from Magentic‑One in terms of **architecture** and **communication**. We **decouple** strategic task decomposition (Planner) from subtask orchestration (Coordinator), and use a **shared task channel** where workers post only concise final results—keeping contexts clean and improving the signal-to-noise ratio for replanning. In contrast, Magentic‑One employs a **single Orchestrator** that centrally plans, tracks, and replans tasks.
>
> Furthermore, the experimental results demonstrate the advantage of our workforce over Magentic-One (workforce with GPT-4o: 60.01 vs. Magentic-One with o1: 46.06). In addition, we conducted an experiment where we merged the Planner and Coordinator into a single agent—similar to Magentic-One—and the result (60.01 vs. 56.36) further highlights the benefit of our decoupled design.
>
>
> **Q5.& W2. Results: if SFT is finetuned on GPT4o-mini, why is there such a significant gap between the first and second to last row in in Table 3.**
>
> Thanks for your valuable feedback. Our **SFT** uses **1,599 filtered trajectories** synthesized by GPT-4o-mini and shows the known pattern of **gains on L1/L2 but a regression on L3 (−3.85%)**, indicating transferring planning style from GPT‑4o‑mini to **Qwen‑32B** is challenging and can overfit to simpler cases. Once we add **DPO**, the planner improves from **41.21% to 52.73%**, **surpassing GPT‑4o‑mini (47.27%)** and yielding **+7.69% on L3**, indicating reinforcement learning is required to learn robust strategies beyond imitation.
>
> **W6. While the approach is promising, it looks like one big limitation is in the necessity to define worker agents per task. The result of this is that, for new tasks, it is still necesary to write agents with domain-specific knowledge. It would be great if authors could comment on the details of how these workers were defined in the experimental section and how scalable this is to new tasks.**
>
>
> The system has the potential to become significantly more scalable through automated agent design. To explore this, we conducted an experiment in which a meta-agent, powered by GPT-4o, automatically generated worker configurations. We tested this approach on 10 diverse tasks, using human-crafted workers as the baseline. Task performance was evaluated using human ratings on a 1–5 scale. The results showed that the meta-agent–generated workers achieved quality scores close to those of manually designed counterparts (average score: 4.3 ± 0.2 vs. 4.7 ± 0.1). We will include the full results in the final version of the paper.
>
> **References**
>
> [1] Pal, Ankit, Logesh Kumar Umapathi, and Malaikannan Sankarasubbu. "Medmcqa: A large-scale multi-subject multi-choice dataset for medical domain question answering." Conference on health, inference, and learning. PMLR, 2022.

---

> > ### Comment · Reviewer_vyod · 2025-08-04
> > **Response**
> >
> > Thanks for the author's responses. They clarified all my doubts. I stand by my score and believe this paper should be accepted to NeurIPS.

---

> > > ### Author Response · Authors · 2025-08-05
> > >
> > > Thank you very much for your thoughtful response and for taking the time to carefully review our rebuttal. We're very glad to hear that our responses were able to address your concerns, and we truly appreciate your support for accepting the paper.

---

### Official Review · Reviewer_rqab · 2025-07-02

**Clarity:** 3
**Significance:** 3
**Originality:** 2
**Rating:** 5
**Confidence:** 4

**Summary:**

The paper proposes Workforce, a multi-agent framework that is modular and scalable, featuring a domain-agnostic planner, a coordinator, and domain-specific worker nodes. The paper also introduces OWL, an optimized workforce learning, an efficient and effective multi-agent training paradigm. Both the proposed framework and the training paradigm yield impressive performance on the GAIA benchmark.

**Questions:**

- Why pass@3 sampling for GPT-4o while pass @1 for Claude-3.7-sonnet?

Also see seaknesses.

**Ethical Concerns:**

["NO or VERY MINOR ethics concerns only"]

**Final Justification:**

I believe with the details and additional experiments added during rebuttal, the paper is worth sharing with the community at the conference. I keep my original recommendation of accept.

**Limitations:**

yes

**Quality:**

3

**Strengths And Weaknesses:**

Strengths

- The performance is impressive.
- The code is fully open-sourced.
- The analysis provides insight for the community.
- The presentation is clear.

Weaknesses

- As a modular framework, there is no ablation study on the contribution of each module. Providing this could make the framework design more convincing. It's now unclear why introducing a coordinator is necessary compared to letting the planner handle this role assignment. Figure 4(c) only discusses the training of the planner and the worker, also gives the impression that the coordinator is not necessary.
- The design choice of using a shared task channel is not supported by experimental results. E.g., would it be even more helpful if the planner could access the context of the worker to better replan?
- Though the centralized mechanism makes the management of context easier, it also limits the potential of the proposed multi-agent system. E.g., workers could not ask for clarification from the higher-level agents (planner here).
- For the goal of generalist multi-agent assistance, it might be worth some discussion on how to select the worker agents, or even creating worker agents autonomously. It's not justified why the three worker agents in 3.2 are chosen for this goal. It seems more of a choice for the GAIA benchmark. Moreover, the generalizability of the trained planner needs to be verified with different worker agent selections if it's supposed to be utilized off-the-shelf by the community.
- For baselines in 3.3, it seems confusing to have (iii) and (iv) as groups, while in Table 1, there are only two groups of (i) (ii).
- It's worth noting that the idea of decoupling strategic planning from domain-specific execution is also explored extensively in the embodied agents literature, which should also be discussed in the paper. [1][2][3]

[1] Building Cooperative Embodied Agents Modularly with Large Language Models. ICLR24

[2] Describe, Explain, Plan and Select: Interactive Planning with Large Language Models Enables Open-World Multi-Task Agents. NeurIPS23

[3] Language models as zero-shot planners: Extracting actionable knowledge for embodied agents. ICML22

---

> ### Author Rebuttal · Authors · 2025-07-31
>
> Thanks for your constructive review.
>
> **W1. As a modular framework, there is no ablation study on the contribution of each module. Providing this could make the framework design more convincing. It's now unclear why introducing a coordinator is necessary compared to letting the planner handle this role assignment. Figure 4(c) only discusses the training of the planner and the worker, also gives the impression that the coordinator is not necessary.**
>
> Thanks for your valuable feedback. Regarding the coordinator, we have two main considerations in the design:
> - **Context growth and decision interference:** We decouple planning from assignment and enforce a shared task channel that surfaces only concise subtask outcomes, so the planner’s input scales with the number of subtasks rather than worker‑log length, preserving strategic reasoning and reliable replanning.
> - **Adaptation to changing worker pools:** The coordinator maintains a capability registry and live state, allowing worker/tool substitutions to be absorbed by registry updates without retraining the planner, thereby retaining a planner‑only policy that generalizes across domains.
>
> Furthermore, we conducted an additional ablation experiment to determine whether to use the coordinator for task assignment. The model uses gpt-4o. The settings of worker and tool are consistent with the second-to-last row in Table 1. The experimental results are the accuracy of GAIA pass@3. The experimental results are as follows:
>
>
> | Method | Level1 | Level2 | Level3 | Avg.|
> | -------- | -------- | -------- | -------- | -------- |
> | **Workforce** | 81.14 | 58.14 | 26.92 | 60.61 |
> | **Workforce wo. Coordinator** | 75.47 | 54.65  | 23.07 | 56.36 |
>
> The experimental results show that not using the coordinator will slightly reduce the performance of the overall system, which confirms our experience.
>
>
> **W2. The design choice of using a shared task channel is not supported by experimental results. E.g., would it be even more helpful if the planner could access the context of the worker to better replan?**
>
> Thanks for your valuable feedback.
>
> (1) Keeping information concise and focused is also critical for enhancing agent performance. One of the key motivations behind using a shared task channel is to strike a balance between retaining essential subtask outcomes and avoiding excessively long contexts. While giving the planner full access to each worker's task context during replanning might provide more comprehensive information, it often introduces a large amount of irrelevant content (e.g., long passages from multiple webpages), which can overwhelm the planner and impair its decision-making.
>
> (2) We conducted an experiment comparing our default setup with a variant where the full trajectory was placed into the shared task channel. The final performance in this setting dropped to 46.06, with most failures attributable to "out of context" issues caused by the noise and overload in the shared task history.
>
>
>
> **W3. Though the centralized mechanism makes the management of context easier, it also limits the potential of the proposed multi-agent system. E.g., workers could not ask for clarification from the higher-level agents (planner here).**
>
> Thanks for your insights. Currently, workers do have a certain ability to provide feedback to the planner. Specifically, when a worker fails in its attempt to solve a downstream task, it generates failure information, including the cause of the failure and possible alternative solutions. This information is then incorporated into the planner's replanning context to assist the planner in replanning.
>
> **W4. For the goal of generalist multi-agent assistance, it might be worth some discussion on how to select the worker agents, or even creating worker agents autonomously. It's not justified why the three worker agents in 3.2 are chosen for this goal. It seems more of a choice for the GAIA benchmark. Moreover, the generalizability of the trained planner needs to be verified with different worker agent selections if it's supposed to be utilized off-the-shelf by the community.**
>
> Thanks for your valuable insights.
>
> (1) Firstly, our worker design follows **two principles**: (1) The overlap in capabilities between different workers should be minimized, and (2) the boundaries of worker capabilities should be clearly defined to facilitate decision-making by the coordinator and planner. We designed our workers based on the sveral capabilities required for general-purpose tasks: web browsing, multimodal processing, file handling, code writing, and reasoning capabilities. Based on this, we designed three workers (Web Agent, Document Processing Agent, and Reasoning/Coding Agent). Of course, more workers can be added to complete other types of tasks, such as adding an terminal agent to handle OS-related tasks.
>
> (2) Secondly, to verify the generalization of the trained planner, we conducted a supplementary experiment: while retaining the GAIA worker configuration, we added an additional agent for medical scenarios. We used the dev dataset from the MedMCQA[1] dataset and tested the planner before and after training (based on the Qwen2.5-32b-Instruct model). The experimental results showed an accuracy of 71% before training and 75% after training, demonstrating that the trained planner maintains generalization capabilities in out-of-domain scenarios.
>
> (3) Regarding automated agent design, we carried out an experiment where a meta-agent, powered by GPT-4o, autonomously generated worker configurations. This setup was evaluated across 10 varied tasks, with manually designed workers serving as the benchmark. Human evaluators rated task performance on a 1–5 scale. The outcome demonstrated that the quality of meta-agent–generated workers was comparable to that of human-designed ones, achieving an average score of 4.3 ± 0.2 versus 4.7 ± 0.1. We plan to include the complete results in the final version of the paper.
>
>
> **W5. For baselines in 3.3, it seems confusing to have (iii) and (iv) as groups, while in Table 1, there are only two groups of (i) (ii).**
>
> We appreciate your detailed insights. One common variant of agent systems is the toolkit-based design. The reason we separated (iii) and (iv) as distinct baselines is that these two were implemented by us with carefully aligned tool sets, allowing for a more controlled and fair comparison focused specifically on agent architecture design. We have clarified this point in the paper (Lines 145–146).
>
> **W6. It's worth noting that the idea of decoupling strategic planning from domain-specific execution is also explored extensively in the embodied agents literature, which should also be discussed in the paper.**
>
> Thanks for your constructive suggestion. We will add more comparisons of OWL and embodied literatures in the related work section of our paper.
>
> **Q1. Why pass@3 sampling for GPT-4o while pass @1 for Claude-3.7-sonnet?**
>
> Thank you for your question. (1) Since GAIA operates in real-world environments (e.g., retrieving information from the web), it is more susceptible to issues such as network errors; thus, we adopted pass\@3 for greater robustness. (2) Due to the high cost of the Claude-3.7-SONNET API, we used pass\@1 for evaluation to manage computational expenses.
>
>
> **References**
>
> [1] Pal, Ankit, Logesh Kumar Umapathi, and Malaikannan Sankarasubbu. "Medmcqa: A large-scale multi-subject multi-choice dataset for medical domain question answering." Conference on health, inference, and learning. PMLR, 2022.

---

> > ### Comment · Reviewer_rqab · 2025-08-06
> > **Thanks for the rebuttal**
> >
> > Thanks for the detailed rebuttal and additional experiments. I believe with these improvements, the paper is worth sharing with the community at the conference.

---

> > > ### Author Response · Authors · 2025-08-07
> > >
> > > Thank you for your thoughtful follow-up and positive assessment. We truly appreciate your time and feedback throughout the review process.

---

### Official Review · Reviewer_RWGZ · 2025-07-03

**Clarity:** 3
**Significance:** 3
**Originality:** 3
**Rating:** 5
**Confidence:** 3

**Summary:**

This paper introduces WORKFORCE, a hierarchical multi-agent framework designed to address the domain-specific limitations of existing LLM-based systems. By decoupling strategic planning from domain-specific execution through a modular architecture (Planner, Coordinator, Worker Nodes), WORKFORCE enables cross-domain transferability with minimal retraining. The framework is validated on the GAIA benchmark, achieving state-of-the-art open-source performance and outperforming commercial systems like OpenAI’s Deep Research. Additionally, the Optimized Workforce Learning (OWL) training paradigm enhances generalization via SFT and reinforcement learning, boosting the Qwen2.5-32B model by 16.37% and demonstrating performance comparable to GPT-4o on complex tasks.

**Questions:**

+ The writing style resembles an engineering report rather than an academic paper, with somewhat simplistic descriptions of algorithmic process details.

+ As shown in Fig. 4, the effectiveness of OWL is primarily constrained by tool capabilities, even more so than the planner. However, tool invocation requires meticulous manual authoring. To fully achieve automation, how can Worker nodes proactively modify and expand their capabilities? This is mentioned in line 107 of the paper, but specific details are absent.

+ Line 102 states, "Planner Agent analyzes incoming tasks and decomposes them into subtasks based on worker capability registry." Evidently, the Planner Agent relies on existing Workers for task decomposition. Thus, when Workers are modified, does the Planner need to be retrained? Can a fine-tuned Planner dynamically adapt to different Worker identities?

**Ethical Concerns:**

["NO or VERY MINOR ethics concerns only"]

**Final Justification:**

In the rebuttal phase, the author addressed my concerns about the algorithm's scalability and generalization. So I increased my score accordingly.

**Limitations:**

yes

**Quality:**

4

**Strengths And Weaknesses:**

**Strengths**

- Clear and excellent academic writing with well-designed figures
- The flexible and modular multi-agent architecture is crucial for swarm intelligence system applications
- Only requiring tuning of the planner reduces training overhead significantly
- State-of-the-art (SOTA) experimental results outperform current renowned frameworks
- Full open-source release of the code


**Weaknesses**

See `Questions`.

---

> ### Author Rebuttal · Authors · 2025-07-31
>
> We appreciate your valuable feedback, and address your concerns as follows:
>
> **Q1. The writing style resembles an engineering report rather than an academic paper, with somewhat simplistic descriptions of algorithmic process details.**
>
> Thank you for your feedback. As this paper proposes both an agent framework and a training method, some content has been simplified due to page limitations. Additional details are provided in the appendix. In the final version, we will further refine this section to enhance its academic rigor, including adding formal definitions for the workforce and OWL components.
>
> **Q2. As shown in Fig. 4, the effectiveness of OWL is primarily constrained by tool capabilities, even more so than the planner. However, tool invocation requires meticulous manual authoring. To fully achieve automation, how can Worker nodes proactively modify and expand their capabilities? This is mentioned in line 107 of the paper, but specific details are absent.**
>
> We sincerely appreciate your insightful comments.
>
> (1) Regarding tool capability expansion, while our current focus is on architectural design, we recognize the importance of automated tool modification. The workforce system can already extend its capabilities through dynamic tool generation – for instance, our agent system automatically created a tool using GitHub API to solve a specific task "According to GitHub, when was Regression added to the oldest closed numpy.polynomial issue that has the Regression label in MM/DD/YY?".
>
> (2) To further demonstrate the potential for automating worker design, we conducted an experiment in which a meta-agent powered by GPT-4o automatically generated worker designs. We evaluated this setup on 10 diverse tasks, using human-designed workers as the baseline for comparison. Quality was assessed using human ratings on a 1–5 scale in terms of prompt/toolkit. The results showed that the workers generated by the meta-agent achieved performance close to that of human-designed ones (average score: 4.3 ± 0.2 vs. 4.7 ± 0.1). We will include these results in the final version.
>
> **Q3. Line 102 states, "Planner Agent analyzes incoming tasks and decomposes them into subtasks based on worker capability registry." Evidently, the Planner Agent relies on existing Workers for task decomposition. Thus, when Workers are modified, does the Planner need to be retrained? Can a fine-tuned Planner dynamically adapt to different Worker identities?**
>
> Thanks for your insightful comments. To verify the ability of generaliation, we conducted a supplementary experiment on **medical** domain, which is not covered in the training curriculum: while retaining the GAIA worker configuration, we added an additional agent for medical scenarios. We used the dev set from the **MedMCQA** benchmark (a question answering benchmark in the medical domain) and tested the planner before and after training (based on the Qwen2.5-32b-Instruct model). The experimental results showed an accuracy of **71% before training** and **75% after training**, demonstrating that the trained planner maintains **generalization capabilities in out-of-domain scenarios**.
>
> **References**
>
> [1] Pal, Ankit, Logesh Kumar Umapathi, and Malaikannan Sankarasubbu. "Medmcqa: A large-scale multi-subject multi-choice dataset for medical domain question answering." Conference on health, inference, and learning. PMLR, 2022.

---

> > ### Comment · Reviewer_RWGZ · 2025-08-03
> >
> > Thanks for the reply. The authors' reply solved all my doubts so I will increase my score from 4 to 5.

---

> > > ### Author Response · Authors · 2025-08-04
> > >
> > > Thank you very much for your thoughtful review and for increasing the score! If you have any further suggestions or feedback, we’d greatly appreciate it.

---

### Official Review · Reviewer_vpYJ · 2025-07-03

**Clarity:** 4
**Significance:** 3
**Originality:** 3
**Rating:** 5
**Confidence:** 4

**Summary:**

This paper introduces a hierarchical multi-agent framework that decouples strategic planning from domain-specific execution to enable cross-domain transferability. The system comprises three core components: a domain-agnostic Planner for task decomposition, a Coordinator for subtask management, and specialized Workers with domain-specific capabilities. The authors also propose Optimized Workforce Learning (OWL), a training paradigm that focuses on optimizing the domain-agnostic planner through supervised fine-tuning followed by reinforcement learning. The approach is evaluated on the GAIA benchmark, achieving 69.70% accuracy and outperforming commercial systems like OpenAI's Deep Research by 2.34%.

**Questions:**

0. Why are you using JSON agent instead of Code agent?

1. How does the system perform on domains that are significantly different from those in the training curriculum? Can you provide analysis on out-of-domain generalization?

2. How does the system handle cases where multiple workers could potentially handle a subtask? What mechanisms prevent conflicts or redundant work?

3. While you mention replanning mechanisms, how robust is the system to cascading failures where multiple workers fail sequentially?

4. What is the computational overhead of the coordinator and communication mechanisms compared to direct single-agent approaches?

5. How does performance degrade as you add more workers or increase task complexity? Are there theoretical or empirical limits to the approach?

**Ethical Concerns:**

["NO or VERY MINOR ethics concerns only"]

**Final Justification:**

My concerns have been addressed. I maintain the original score.

**Limitations:**

Yes

**Quality:**

3

**Strengths And Weaknesses:**

## Strengths:

- Novel Modular Architecture: The separation of domain-agnostic planning from domain-specific execution is well-motivated and represents a significant architectural innovation. This design enables plug-and-play extensibility across domains without requiring complete system redesign.

- Strong Empirical Results: The paper demonstrates impressive performance on GAIA, achieving state-of-the-art results among open-source frameworks (69.70%) and surpassing proprietary systems like OpenAI's Deep Research. The OWL training method shows substantial improvements (+16.37% for Qwen2.5-32B-Instruct).

- Comprehensive Evaluation: The paper includes thorough ablation studies, error analysis, and performance breakdowns across different capability types. The test-time scaling analysis and robustness evaluation add valuable insights.

- Reproducibility: The authors commit to releasing all code, models, and data, which supports open research.


## Weaknesses

- Tool Dependency: The system's performance is heavily dependent on the quality and availability of domain-specific tools. The paper doesn't adequately address how to handle domains with poor tooling.

- Scalability Concerns: The paper doesn't thoroughly address how the approach scales with the number of workers or the complexity of coordination required for very large multi-agent systems.

Missed References:

[1] Multi-modal Agent Tuning: Building a VLM-Driven Agent for Efficient Tool Usage.

[2] From Exploration to Mastery: Enabling LLMs to Master Tools via Self-Driven Interactions.

---

> ### Author Rebuttal · Authors · 2025-07-31
>
> Thanks for your valuable comments. We will answer the following questions in detail one by one.
>
> **W1. The system's performance is heavily dependent on the quality and availability of domain-specific tools. The paper doesn't adequately address how to handle domains with poor tooling.**
>
> We appreciate your thoughtful insights. We admit that tool design does require careful manual design. However, this paper's focus is mainly on modular **agent architecture**. One solution to this problem is to let the agent generate the tool itself, which we plan to explore in the future work.
>
> Furthermore, one emergent behavior of the current system is that, when faced with tasks that the current tool set cannot handle—for example, "According to GitHub, when was Regression added to the oldest closed numpy.polynomial issue that has the Regression label in MM/DD/YY?"—our system used the **code execution tool** to **write a function leveraging the GitHub API**, and correctly answered the question.
>
> **W2&Q6. Scalability Concerns: The paper doesn't thoroughly address how the approach scales with the number of workers or the complexity of coordination required for very large multi-agent systems. How does performance degrade as you add more workers or increase task complexity? Are there theoretical or empirical limits to the approach?**
>
> Thanks for your insightful consideration. Existing figures show **robustness on multi‑capability tasks** (Figure. 4 (b)) and **positive test‑time scaling** (Figure 3(b)).
> Besides, We believe that the key here is whether the system can correctly identify which worker a subtask under the current task domain should be assigned to. In light of this, we conducted additional experiments on PMC-VQA[1] and PathVQA[2], two medical VQA benchmarks. We first use all default sub-agents: `reasoning_coding_agent`, `document_processing_agent`, and `web_agent`. Besides, to better handle image-related questions, we also include an `image_analysis_agent`. Here are the experimental results:
>
> PMC-VQA Results:
> | Model | Accuracy |
> |-------|----------|
> | Claude-3.7-Sonnet | 77% |
> | GPT-4O | 74% |
>
> PathVQA Results:
> | Model | F1 Score | Exact Match |
> |-------|----------|-------------|
> | Claude-3.7-Sonnet | 59.2 | 18.9 |
> | GPT-4O | 52.3 | 14.9 |
>
> We observe that our multi-agents system typically only use `image_analysis_agent` and `reasoning_coding_agent` (only for output answer) in its planning, demonstrating planner's ability to adaptively select relevant domain-specific agents based on task requirements when increasing workers, showcasing its flexibility in handling domain-specific challenges.
>
> We will add more analysis such as consistently scaling up worker numbers with accuracy/latency/cost curves in our final version of the paper.
>
> **Q1. Why are you using JSON agent instead of Code agent?**
>
> Thanks for your valuable consideration. We will answer your questions from the following four points：
>
> 1. While OWL is designed as a **general‑purpose agentic system**, executable‑code agents (e.g., CodeAct) — though strong on data/algorithm‑heavy subtasks — are **not universally superior** to JSON/function‑calling across diverse tasks.
> 2. CodeAct’s reference setup spins up a Docker‑isolated Jupyter kernel per session—good for isolation, but it introduces **cold‑start latency** and **non‑trivial CPU/memory/storage overhead**, increases governance/ops burden, and makes outcomes sensitive to **environment drift** (runtime/deps/network).
> 3. Meanwhile, the community’s dominant practice—and much instruction data for base models—centers on **structured JSON/tool calls**, so models are typically better calibrated to that protocol.
> 4. Thanks to OWL’s **modular design**, a CodeAct‑style execution path can be added as an **optional Worker** without changing the architecture, allowing us to **benefit from both** when appropriate.
>
> **Q2. How does the system perform on domains that are significantly different from those in the training curriculum? Can you provide analysis on out-of-domain generalization?**
>
> Thanks for your comments. Firstly, there's already a gap between our training curriculum and GAIA. For example, our training curriculum doesn't include training data related to **file processing** for Excel, Docx, and Python, whereas many GAIA use cases require the agent to perform file processing. Secondly, to verify the ability of generaliation, we conducted a supplementary experiment on **medical** domain, which is not covered in the training curriculum: We added an additional agent for medical scenarios based on the default worker set configuration, to test whether planner can handle medical tasks effectively. We used the dev set from the **MedMCQA**[3] benchmark (a question answering benchmark in the medical domain) and tested the planner before and after training (based on the Qwen2.5-32b-Instruct model). The experimental results showed an accuracy of **71% before training** and **75% after training**, demonstrating that the trained planner maintains **generalization capabilities in out-of-domain scenarios**.
>
> **Q3. How does the system handle cases where multiple workers could potentially handle a subtask? What mechanisms prevent conflicts or redundant work?**
>
> When multiple workers can handle a subtask, the **Coordinator** makes the unified decision about assigning the subtask. **Worker descriptions** (including capability boundaries) are crucial for the coordinator's decision-making (currently manually constructed). If a worker mistakenly assigns a subtask to a worker that lacks the required capabilities, a **test-time replanning** mechanism kicks in. The planner receives the reason for the downstream subtask failure and attempts to replan the subtasks to align them as closely as possible with the worker's capabilities.
>
> **Q4. While you mention replanning mechanisms, how robust is the system to cascading failures where multiple workers fail sequentially?**
>
> Thanks for your insightful feedback. Workers perform self‑assessment and post structured failure reports (reasons and alternatives) to the shared channel, and then the planner re‑plans based on this feedback. This enables the system to identify issues earlier—rather than waiting until final execution to debug—thereby reducing cross‑worker failure propagation. In addition, Figure 3 shows that performance improves as the number of replanning iterations increases from 0 to 2—for example, the GPT‑4o workforce rises from **0.43** to **0.60**.
>
> **Q5. What is the computational overhead of the coordinator and communication mechanisms compared to direct single-agent approaches?**
>
> When starting a workforce to try to solve a task in GAIA, the total average cost of the coordinator is 3458.89 tokens. The coordinator consumes an average of 1033.53 tokens to assign a worker. Maintaining the task channel involves some communication, consuming an average of 734 tokens.
>
> **References**
>
> [1] Zhang, Xiaoman, et al. "Pmc-vqa: Visual instruction tuning for medical visual question answering." arXiv preprint arXiv:2305.10415 (2023).
>
> [2] He, Xuehai, et al. "Pathvqa: 30000+ questions for medical visual question answering." arXiv preprint arXiv:2003.10286 (2020).
>
> [3] Pal, Ankit, Logesh Kumar Umapathi, and Malaikannan Sankarasubbu. "Medmcqa: A large-scale multi-subject multi-choice dataset for medical domain question answering." Conference on health, inference, and learning. PMLR, 2022.

---

> > ### Comment · Reviewer_vpYJ · 2025-08-05
> > **Response to Rebuttal**
> >
> > Thank the authors for the detailed rebuttal. My concerns have been addressed. I believe the added experiments, discussions, and references meaningfully strengthen the paper's contributions and practical relevance. Since I have already given a positive score, I will maintain my positive evaluation.

---

> > > ### Author Response · Authors · 2025-08-05
> > >
> > > Thank you for your thoughtful follow-up and for taking the time to carefully consider our rebuttal. We truly appreciate your constructive engagement!

---

### Decision · Program_Chairs · 2025-09-17

**Decision:**

Accept (poster)

**Comment:**

All the reviewers agree to accept this paper, so I recommend acceptance as well. However, I don't think this work is significantly novel. Re-training planner alone is not unique in this work. Issues such as scalability along the subtasks are not handled, as pointed out by reviewers and admitted by the authors.  To be frank, I've found myself learned nothing after reading this paper.